EMBO
Molecular Medicine

# Constitutive hippocampal cholesterol loss underlies poor cognition in old rodents

Mauricio G Martin[1,2,*], Tariq Ahmed[3], Alejandra Korovaichuk[4], Cesar Venero[5], Silvia A Menchón[2,6], Isabel Salas[1], Sebastian Munck[2], Oscar Herreras[4], Detlef Balschun[3] & Carlos G Dotti[1,2,**]

## Abstract

Cognitive decline is one of the many characteristics of aging. Reduced long-term potentiation (LTP) and long-term depression (LTD) are thought to be responsible for this decline, although the precise mechanisms underlying LTP and LTD dampening in the old remain unclear. We previously showed that aging is accompanied by the loss of cholesterol from the hippocampus, which leads to PI3K/Akt phosphorylation. Given that Akt de-phosphorylation is required for glutamate receptor internalization and LTD, we hypothesized that the decrease in cholesterol in neuronal membranes may contribute to the deficits in LTD typical of aging. Here, we show that cholesterol loss triggers p-Akt accumulation, which in turn perturbs the normal cellular and molecular responses induced by LTD, such as impaired AMPA receptor internalization and its reduced lateral diffusion. Electrophysiology recordings in brain slices of old mice and in anesthetized elderly rats demonstrate that the reduced hippocampal LTD associated with age can be rescued by cholesterol perfusion. Accordingly, cholesterol replenishment in aging animals improves hippocampal-dependent learning and memory in the water maze test.

**Keywords** aging; cholesterol; learning; LTD; PI3K

**Subject Categories** Aging; Metabolism; Neuroscience

## Introduction

Aging is associated with cognitive decline, such that individuals older than 65 develop cognitive deficits or age-associated memory impairments. The hippocampus, a brain structure that is central to the formation of declarative and other types of memory, is particularly sensitive to aging. However, these impairments are not paralleled by an increase in neuronal death (Burke & Barnes, 2006), indicative that more subtle mechanisms must be affected by aging to produce memory decline.

We previously showed that the phosphorylated form of the serine–threonine kinase Akt (p-Akt) accumulates in old hippocampal neurons, both *in vivo* and *in vitro* (Martin *et al*, 2008, 2011; Trovò *et al*, 2013), which probably reflects the strong need for survival signaling. Nevertheless, the counter effect of robust survival might be reduced performance. In fact, p-Akt phosphorylation negatively regulates long-term depression (LTD), a key process in learning and memory (see below and Peineau *et al*, 2007).

The accumulation of p-Akt with age may be the result of different processes. We previously observed that the increase with age of the cholesterol hydroxylating enzyme cholesterol 24-hydroxylase (CYP46A1), the major catabolic enzyme of cholesterol in the brain (Lund *et al*, 2003), triggers cholesterol loss-dependent, ligand-independent, activation of the TrkB receptor and, consequently, Akt phosphorylation (Martin *et al*, 2008, 2011; Sodero *et al*, 2011a,b). A second mechanism driving the increase in p-Akt with age is also linked to cholesterol loss, and it involves the increase in plasma membrane sphingomyelin, similarly contributing to ligand-independent TrkB phosphorylation and PI3K/Akt activation (Trovò *et al*, 2011). Increased PI3K/Akt activity in old neurons also seems to arise from the constitutive increase with age of interleukin 1B (IL-1B) activity and from the age-associated detachment from the plasma membrane of the PIP2 binding protein, myristoylated alanine-rich C kinase substrate (MARCKS: Trovò *et al*, 2013). Altered binding of MARCKS to the membrane favors the accumulation of PIP3 in the synaptic fraction of old mice, with the subsequent increase in Akt phosphorylation (Trovò *et al*, 2013).

Activity-dependent changes in synaptic strength, such as long-term potentiation (LTP) and long-term depression (LTD) of synaptic transmission, are considered to be the cellular substrates of learning and memory (Neves *et al*, 2008; Collingridge *et al*, 2010). Competitive interactions between these forms of synaptic plasticity have

1 Centro Biología Molecular "Severo Ochoa" CSIC-UAM, Madrid, Spain
2 VIB Center for the Biology of Disease, Center for Human Genetics, University of Leuven (KU Leuven), Leuven, Belgium
3 Laboratory of Biological Psychology, Faculty of Psychology and Educational Sciences, University of Leuven (KU Leuven), Leuven, Belgium
4 Departamento de Neurobiología Funcional y de Sistemas, Instituto Cajal – CSIC, Madrid, Spain
5 Departamento de Psicobiología, Facultad de Psicología, UNED, Madrid, Spain
6 IFEG-CONICET and FaMAF, Universidad Nacional de Córdoba, Córdoba, Argentina
  *Corresponding author. Tel: +54 351 4681465; E-mail: mauricio.m@immf.uncor.edu
  **Corresponding author. Tel: +34 911964401; E-mail: cdotti@cbm.csic.es

been proposed to participate in memory storage (Diamond *et al*, 2005; Nicholls *et al*, 2008; Ge *et al*, 2010). Indeed, glycogen synthase kinase 3 beta (GSK3β) determines whether NMDA receptor activation induces or inhibits LTD (Peineau *et al*, 2007). Moreover, the activation of protein phosphatase 1 (PP1) during LTD dephosphorylates GSK3β at Ser9, resulting in its activation and receptor internalization. Since p-Akt is a GSK3β inhibitor, PP1-mediated Akt dephosphorylation further contributes to the enhancement of GSK3β activity. Significantly, GSK3β and Akt dephosphorylation are impaired by okadaic acid, a compound that also blocks LTD by inhibiting PP1 (Peineau *et al*, 2007). Conversely, activation of NMDA receptors leads to the stimulation of PI3K/Akt pathway during LTP, provoking the phosphorylation of GSK3β at Ser9 to prevent LTD. Thus, it appears reasonable to assume that the loss of cholesterol and the consequent enhancement of PI3K/Akt activity that occurs in the hippocampus during aging will have a detrimental effect on LTD and, consequently, on cognition. Since most studies coincide that aging is accompanied by a decreased propensity to develop LTD (Lee *et al*, 2005; Billard & Rouaud, 2007; Ahmed *et al*, 2011), we investigated here whether and how this is related to age-associated constitutive cholesterol loss.

## Results

### Impaired LTD-induced Akt dephosphorylation in the aged hippocampus

The magnitude of the LTD response seems to depend on the efficacy of AMPAR internalization, which is in turn dependent on Akt dephosphorylation, GSK3β activation, and receptor endocytosis. Accordingly, in an aged hippocampus in which the equilibrium is displaced toward pAkt (see Introduction), incomplete pAkt dephosphorylation would result in a poor LTD response. To test this prediction, p-Akt levels were measured in hippocampal slices of young (4 month old; 4M) and old (20 month old; 20M) mice, in unstimulated control conditions, and after pharmacological induction of LTD. In control conditions, there was $1.89 \pm 0.298$ fold more p-Akt in old than in young mice ($P = 0.041$, Fig 1A), and moreover, NMDA triggered a significant dephosphorylation of p-Akt in hippocampal slices from young but not old mice (Fig 1A). In control experiments, the normalization of Akt to β-actin showed that there was no change in total Akt following stimulation and that the amount of p-Akt/β-actin only decreased in young mice after NMDA stimulation (Fig 1A).

To test the possibility that the reduced capacity to dephosphorylate Akt upon stimulation of the aged hippocampus is related to the loss of cholesterol in the synaptic fraction (Martin *et al*, 2008; Sodero *et al*, 2011a,b), we replenished cholesterol in the membranes of hippocampal slices from 20M mice. Exposing slices to 30 μM cholesterol-methyl-β-cyclodextrin (cholesterol-MβCD) and 5 μM cholesterol for 60 min reduced the levels of p-Akt to $47\% \pm 10.3$ of the 20M controls, levels similar to those found in slices from young mice (Fig 1B). To ascertain whether this replenishment strategy did in fact restore cholesterol in the plasma membrane, the levels of this sterol were measured in hippocampal membrane fractions from young and old mice, before and after exposure to the cholesterol-MβCD/cholesterol mix. The cholesterol: total protein ratio was $0.420 \pm 0.053$ in hippocampal membranes

from 4M mice and $0.326 \pm 0.044$ in hippocampal membranes from 20M mice, confirming a loss of more than 20% of the cholesterol with age (see Martin *et al*, 2008 and Sodero *et al*, 2011a,b). After replenishment, the cholesterol levels in membranes from 20M mice increased the cholesterol:protein ratio to 0.398 ($\pm 0.070$ μg), representing a recovery of 17.12% and a replenishment to 95% the levels in hippocampal membranes from young mice.

In a control experiment, we tested the effect of adding cholesterol to slices prepared from young mice, and in 4M acute hippocampal slices, this did not alter the amount of p-Akt or Akt in basal conditions (Fig 1C). Moreover, NMDA triggered p-Akt dephosphorylation in both control and cholesterol-replenished slices, maintaining similar levels of p-Akt after stimulation in both cases (4M control + NMDA: $75 \pm 8\%$; 4M cholesterol + NMDA: $81 \pm 5\%$; $P = 0.694$).

We recently demonstrated that in old mice, hippocampal synapses contain less MARCKS in membranes, resulting in reduced PI(4,5)P2 clustering and increased levels of PI(3,4,5)P3 and p-Akt (Trovò *et al*, 2013). To gain further insight into how cholesterol loss during aging may increase pAkt, we investigated the relationship between cholesterol loss and the association of MARCKS with the plasma membrane. When hippocampal neurons were treated with a low dose of the cholesterol-extracting drug MβCD, there was a $23.63 \pm 5.60\%$ loss of cholesterol and a significant reduction in membrane-bound MARCKS (Fig 1D). This reduction of MARCKS in low-cholesterol membranes was not due to protein degradation, as analyzing the membrane and supernatant fractions from control and cholesterol-depleted neurons in Western blots demonstrated that MβCD treatment provoked MARCKS detachment from the membranes with its concomitant increase in the soluble fraction (Fig 1D). Finally, we assessed whether the addition of cholesterol to hippocampal slices prepared from young mice affected the distribution of MARCKS. Accordingly, we found that the addition of cholesterol to acute hippocampal slices from 4M mice did not affect the levels of MARCKS in the membrane fractions of those slices (Fig 1E).

To demonstrate a functional connection between the reduction in MARCKS at the membrane and Akt phosphorylation, the amount of p-Akt was measured in 15 DIV neurons infected with lentiviral particles that express a shRNA designed to knockdown MARCKS expression (shMARCKS). MARCKS knockdown was associated with increased p-Akt levels (Supplementary Fig S1), and the same effect observed following the overexpression of a mutant form of MARCKS that cannot bind PI(4,5)P2 (Trovò *et al*, 2013). When cholesterol was extracted from shMARCKS-treated neurons, a further increase of Akt phosphorylation was observed, supporting our hypothesis that cholesterol loss and membrane MARCKS detachment both contribute to Akt phosphorylation (Supplementary Fig S1).

The phosphatase and tensin homolog deleted on chromosome ten (PTEN) catalyzes the conversion of PI(3,4,5)P3 to PI(4,5)P2. It was shown previously that during NMDA-induced LTD, PTEN is recruited to the postsynaptic density (PSD) and that this is a strict requirement for this type of LTD (Jurado *et al*, 2010). Hence, we tested whether impaired p-Akt dephosphorylation after LTD in old mice may be also due to deficits in PTEN activity. It might be expected that deficient PI(3,4,5)P3 degradation by PTEN would lead to the subsequent accumulation of PI(3,4,5)P3 and p-Akt hyperactivation. Hence, PTEN levels were analyzed in the PSD fraction of hippocampal slices from 4M and 20M mice, 30 min after NMDA-

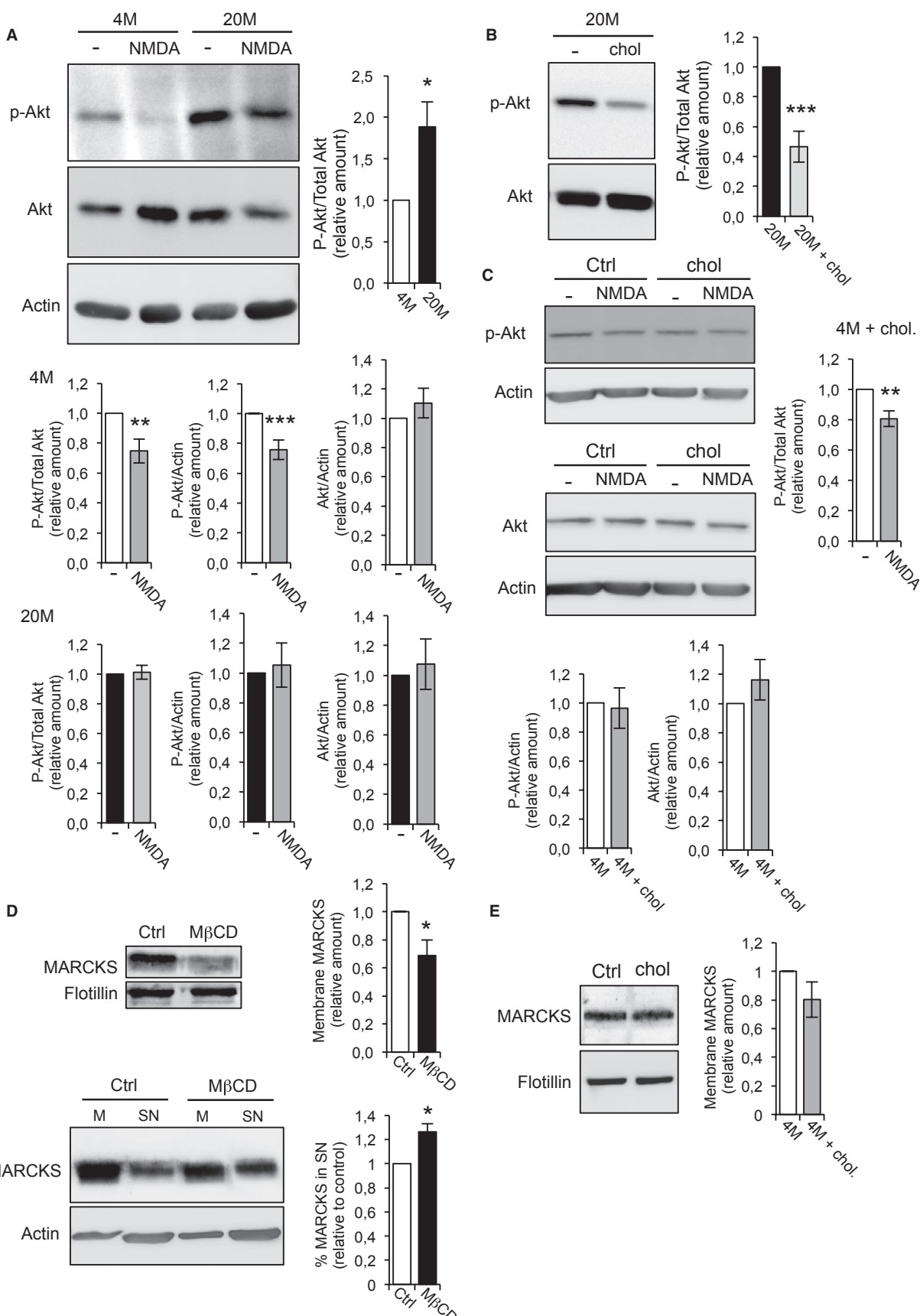

**Figure 1.**

induced LTD. Recruitment of PTEN to the PSD fraction after NMDA-LTD was similar in the membranes from young and old hippocampal slices (Fig 2).

Together, these studies indicated that the loss of cholesterol in the hippocampus of aged mice produces a strong increase in PI3K/pAkt activity. Moreover, MARCKS detachment driven by cholesterol loss, a protein previously shown to be associated with high pAkt levels in the aged (Trovò *et al*, 2013), seems also to be another possible upstream determinant.

**Cholesterol loss results in surface accumulation and impaired endocytosis of AMPARs**

To determine the extent to which the enhanced PI3K/pAkt driven by cholesterol loss in the old affects the basic molecular machinery of cognition, we analyzed AMPAR dynamics in hippocampal neurons *in vitro*. In general, cells *in vitro* are particularly useful to study and quantify receptor dynamics, and hippocampal neurons *in vitro* are particularly suited as, like hippocampal cells *in situ*, they also undergo a significant reduction in plasma membrane cholesterol over time *in vitro*, which in turn is due to the increased expression of the enzyme CYP46A1 (Martin *et al*, 2008). Furthermore, hippocampal neurons also experience a time-dependent increase in p-Akt levels driven by cholesterol loss (Martin *et al*, 2011).

To study AMPAR dynamics, we measured the surface levels and internalization rate of GluA2 containing receptors, since these are the most abundant in the mature hippocampal neurons (Wenthold *et al*, 1996; Malenka, 2003). Significantly more AMPARs were present on the cell surface of 30 DIV neurons compared to 15 DIV neurons (Fig 3A and B), and measurements of the area covered indicated that GluA2-AMPARs were present in larger clusters on 30 DIV neurons (Fig 3C). By contrast, 15 and 30 DIV neurons contained similar amounts of total GluA2 (Supplementary Fig S2A), excluding any possibly differences in expression.

We analyzed the effect of neuronal stimulation on AMPAR endocytosis in 15 and 30 DIV neurons. In the presence of 100 μM glutamate, which induces AMPAR endocytosis in hippocampal cultures

through a mechanism also employed during LTD (Beattie *et al*, 2000; Man *et al*, 2000), weaker surface GluA2 staining was evident in 15 DIV neurons. By contrast, there was no such effect on the number of surface AMPARs in 30 DIV neurons (Fig 3A and B), suggesting that internalization is impaired in the older neurons. Indeed, when the levels of internalized GluA2-AMPARs were measured using an antibody-feeding strategy followed by acid wash (see Materials and Methods), receptor internalization in 15 DIV neurons increased by 33% in the presence of 100 μM but not in 30 DIV neurons (Fig 3D–G). To test whether the reduction in the levels of cholesterol in these older cells might be responsible for the weaker receptor internalization of 30 DIV neurons, these cells were incubated with the cholesterol-MβCD (30 μM) and cholesterol (5 μM) replenishment mix. Under these conditions, there was a significant loss of surface AMPARs on 30 DIV neurons stimulated with glutamate (Fig 4A and B). Indeed, the levels of cholesterol in the neuronal membranes were measured after replenishment, and in 30 DIV hippocampal neurons, the cholesterol-MβCD/cholesterol mix produced a 19% increase in membrane cholesterol: the cholesterol levels in membrane preparations rose from $0.144 \pm 0.002$ to $0.171 \pm 0.011$ μg cholesterol/μg of protein. By contrast, the total levels of GluA2 were not affected by cholesterol replenishment *in vitro* or *in vivo* (Supplementary Figs S2A and S3).

This second series of experiments indicates that the loss of cholesterol with aging may perturb certain aspects of cognition by virtue of defective AMPAR internalization, justifying the reduced LTD typical at this stage of life.

**Cholesterol loss affects the lateral mobility of AMPA receptors**

In order to study how changes in cholesterol levels may affect AMPAR behavior in more detail, we next measured AMPAR lateral diffusion. LTD requires the rapid redistribution of receptors away from the synapse by lateral diffusion (Tardin *et al*, 2003), and the subsequent internalization of the displaced receptors by endocytosis is essential to sustain LTD (Carroll *et al*, 1999; Beattie *et al*, 2000). Mechanistically, it has been proposed that the mobility of synaptic

**Figure 1.  p-AKT dephosphorylation after LTD is impaired in old mice.**

A    p-Akt levels in acute hippocampal slices prepared from young (4M) and old (20M) mice in control conditions or 1 h after NMDA-LTD induction were assessed in Western blots. The quantification shows that in control conditions, the levels of p-Akt are 50% higher in 20M than in 4M mice. Stimulation with 20 μM NMDA induced p-Akt dephosphorylation in 4M but not in 20M mice. The bar plots show the levels of p-Akt in young and old mice, corrected for total Akt and for β-actin, in controls and after stimulation. The quantification of total Akt/β-actin shows that the total Akt levels do not change after stimulation. The levels of p-Akt/Akt after stimulation were: 4M = $0.75 \pm 0.081$ ($n = 5$ animals, $P = 0.014$), and 20M = $1.01 \pm 0.047$, ($n = 5$ animals, $P = 0.767$). The levels of p-Akt/β-actin after stimulation were: 4M = $0.76 \pm 0.065$, ($n = 5$ animals, $P = 0.004$), and 20M = $1.05 \pm 0.15$, ($n = 8$ animals, $P = 0.726$). The levels of total Akt/β-actin after stimulation were: 4M = $1.10 \pm 0.101$ ($n = 5$ animals, $P = 0.340$) and 20M = $1.08 \pm 0.170$, ($n = 8$ animals, $P = 0.664$).

B    Western blot and quantification showing that addition of cholesterol to acute slices from 20M mice restores the basal levels of p-Akt observed in young mice [20M + chol = $0.47 \pm 10.3$ ($n = 5$ animals, $P = 0.0008$)].

C    Control experiments show that the addition of cholesterol to acute hippocampal slices from 4M mice does not effect on the levels of p-Akt and total Akt, or on p-Akt dephosphorylation after 20 μM NMDA stimulation. p-Akt/β-actin: 4M + chol = $0.96 \pm 0.139$ ($n = 5$ animals, $P = 0.805$); Akt/β-actin: 4M + chol = $1.16 \pm 0.138$ ($n = 5$ animals, $P = 0.311$). The levels of p-Akt/Akt after stimulation were: 4M + chol = $0.81 \pm 0.051$ ($n = 5$ animals, $P = 0.0143$).

D    Lower levels of the PIP2-binding protein MARCKS were found in membrane preparations from 15 DIV cholesterol-depleted neurons compared to controls. The plot shows the amount of MARCKS attached to the membrane relative to controls: control neurons = 1.00; MβCD-treated neurons = $0.688 \pm 0.11$ ($P = 0.045$, $n = 5$ different cultures). The Western blots below show the membrane-supernatant distribution of MARCKS in control neurons or after cholesterol depletion by MβCD. As can be seen in the blots, MβCD provokes a reduction in the MARCKS present in the membrane fraction (M) with a concomitant $26.33 \pm 6.4\%$ increase in the supernatant (SN, $P = 0.02$, $n = 3$ different cultures).

E    The Western blot corresponds to control experiments showing that the addition of cholesterol to hippocampal slices from 4M mice does not affect the amount of MARCKS found in membrane fractions: 4M + chol = $0.80 \pm 0.124$ ($P = 0.187$, $n = 3$ animals).

Data information: In (A-E), the presented values are relative to controls, considered as 1. The P-values correspond to 2-sided t-test.
Source data are available for this figure.

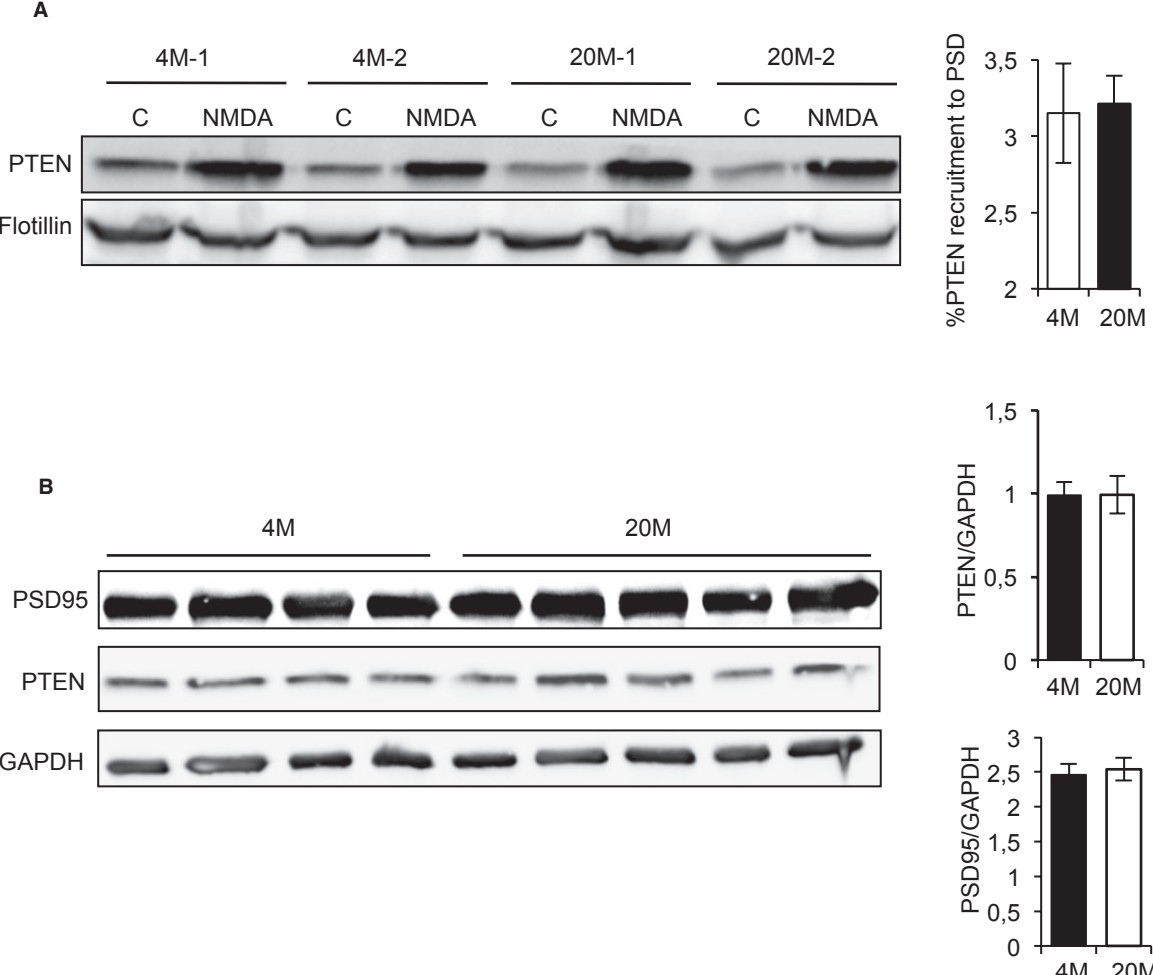

**Figure 2.  Impaired Akt dephosphorylation in old mice is not due to deficits in PTEN.**

A   No differences were observed in PTEN recruitment to the PSD after LTD induction in young and old mice. The Western blot shows the levels of PTEN found in the PSD purified from acute hippocampal slices at 30 min after LTD. No difference in the levels of recruited PTEN was observed between 4- and 20-month-old mice (mean ± standard error, $n$ = 5 animals from each age).

B   Total levels of PSD95 and PTEN do not change comparing young and old mice (mean ± standard error, $n$ = 5 animals from each age).

Source data are available for this figure.

but not extra-synaptic receptors during LTD requires PI(3,4,5)P3 depletion (Arendt *et al*, 2010). Hence, the loss of cholesterol in the older cells would probably affect AMPAR displacement by lateral diffusion, similar to, and possibly as a consequence of reduced internalization (see Figs 3 and 4). To test this prediction, we used quantum-dot-based single molecule tracking (Fig 5 and Supplementary Methods).

The instantaneous diffusion coefficients ($D$) for confined (within synapses, $D_{in}$) and non-confined (beyond synapses, $D_{out}$) AMPARs were 0.023 ± 0.0028 $\mu m^2/s$ and 0.0613 ± 0.0099 $\mu m^2/s$, respectively, in 15 DIV neurons (Fig 5D), similar to those in neurons cultured for 30 DIV: $D_{in}$ = 0.0269 ± 0.0085 $\mu m^2/s$ and $D_{out}$ = 0.086 ± 0.0213 $\mu m^2/s$ (Fig 5E). Control experiments where synapses were stained using the synaptic marker mitotracker (Groc *et al*, 2004) showed that the $D_{in}$ obtained for sites of confinement corresponded to $D_{in}$ particles within synapses (Supporting Information). As opposed to the basal levels, in the presence of 100 $\mu M$

glutamate, there was an increase of the mean $D_{in}$ and $D_{out}$ in 15 DIV neurons (Fig 5D), while glutamate was unable to induce any change in AMPAR mobility in 30 DIV neurons, neither within nor beyond synapses (Fig 5E). To determine whether this reduced mobility in 30 DIV neurons was the consequence of these cells' lower cholesterol content, we repeated these measurements in the presence of the cholesterol-replenishing solution (see above). Cholesterol replenishment did not affect AMPAR diffusion in control conditions, although glutamate stimulation increased the mobility of synaptic AMPARs, increasing the $D_{in}$ value in the cholesterol reinforced cells from 0.0250 ± 0.0035 $\mu m^2/s$ to 0.0337 ± 0.0047 $\mu m^2/s$ ($P$ = 0.0254, Fig 5F).

Since glutamate was used instead of NMDA to stimulate hippocampal neurons in culture, control experiments were performed to compare the effect of NMDA and glutamate on AMPAR lateral diffusion. We observed that the addition of 20 $\mu M$ NMDA to 15 DIV neurons also increased the values of $D_{in}$ and $D_{out}$, and no

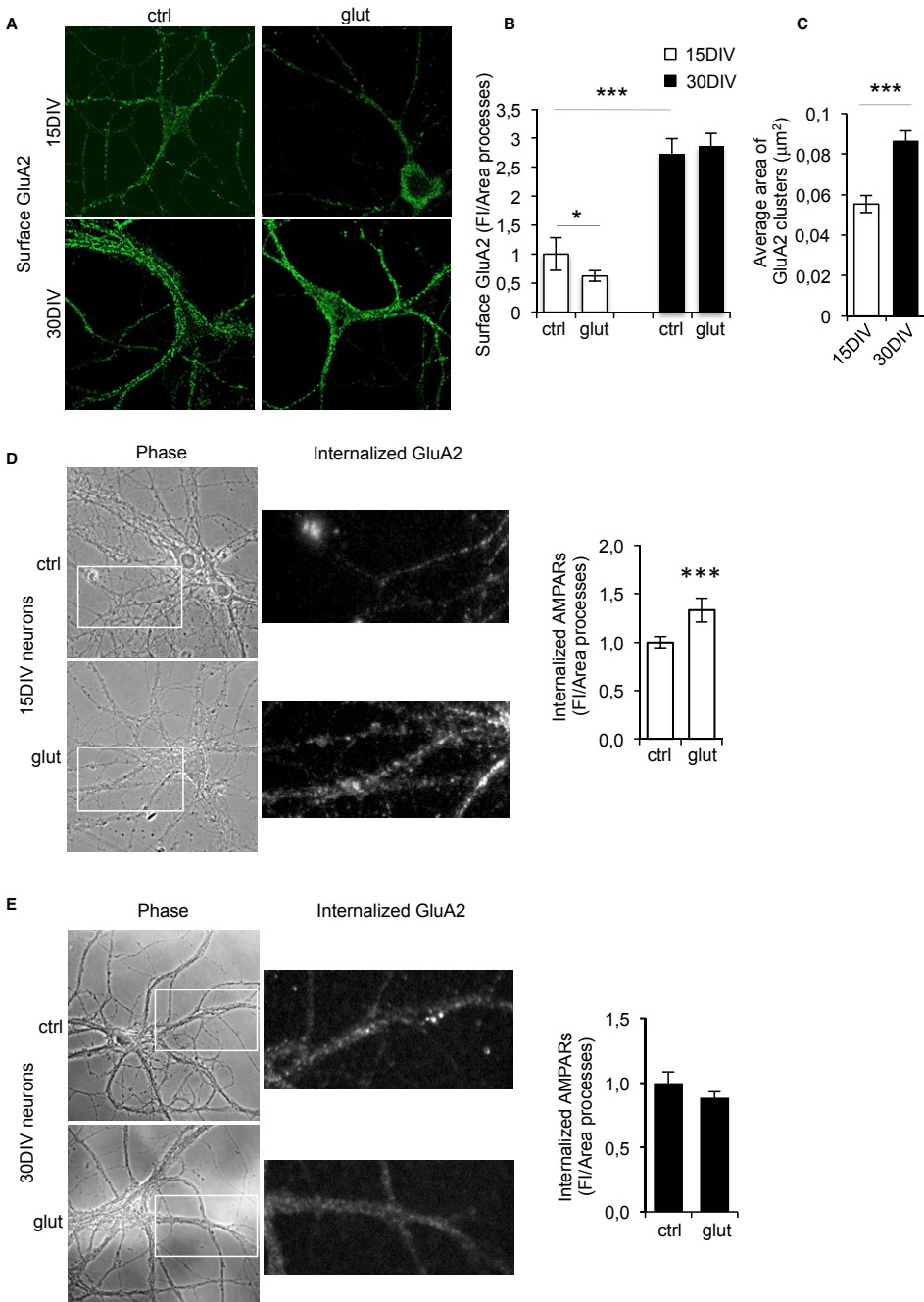

**Figure 3.  Surface GluA2-AMPARs accumulate in low-cholesterol hippocampal neurons.**

A, B    The surface staining of GluA2-containing AMPARs (A) is significantly higher in 30 DIV than in 15 DIV neurons (15 DIV = 1.00 $\pm$ 0.28; 30 DIV = 2.73 $\pm$ 0.27; $P < 0.0001$, $n = 3$). Stimulation with glutamate (100 $\mu$M) resulted in decreased GluA2 staining of 15 DIV (15 DIV glut = 0.62 $\pm$ 0.095; $P^{control, glut}$ = 0.0310; $n = 3$) but not 30 DIV neurons (30 DIV glut = 2.86 $\pm$ 0.219; $P^{control, glut}$ = 0.053; $n = 3$), indicating that low-cholesterol neurons have a reduced capability to endocytose AMPARs in response to ligand. Fluorescence intensity (FI/area) quantified in the processes of neurons is shown in (B).

C    Quantification of the area of GluA2-AMPARs clusters in 15 and 30 DIV neurons indicates that larger receptor clusters form in low-cholesterol neurons (average area 15DIV: 0.055 $\pm$ 0.0042 $\mu$m$^2$, 30 DIV: 0.086 $\pm$ 0.0053 $\mu$m$^2$, $P = 0.0009$, $n = 3$ different cultures).

D, E    Fluorescence images show the internalized AMPARs in 15DIV and 30DIV neurons, before and after glutamate stimulation. The fluorescence images correspond to higher magnifications of the regions indicated in the insets. Glutamate exposure resulted in increased AMPAR internalization in 15 DIV neurons (D) and 5 min after glutamate addition the fluorescence intensity (FI)/area measured in the processes increased from 1 $\pm$ 0.06 to 1.33 $\pm$ 0.12 ($t$-test, $P = 0.0082$, $n = 4$). Glutamate addition did not increase AMPAR internalization in 30 DIV neurons (E). The values of FI/area in the processes were 1 $\pm$ 0.082 and 0.88 $\pm$ 0.048 ($t$-test, $P = 0.252$, $n = 3$).

Data information: The values represent the mean $\pm$ s.e.m. relative to controls. $n$: number of different experiments. The data were compared using Mann-Whitney non-parametric $t$-test.

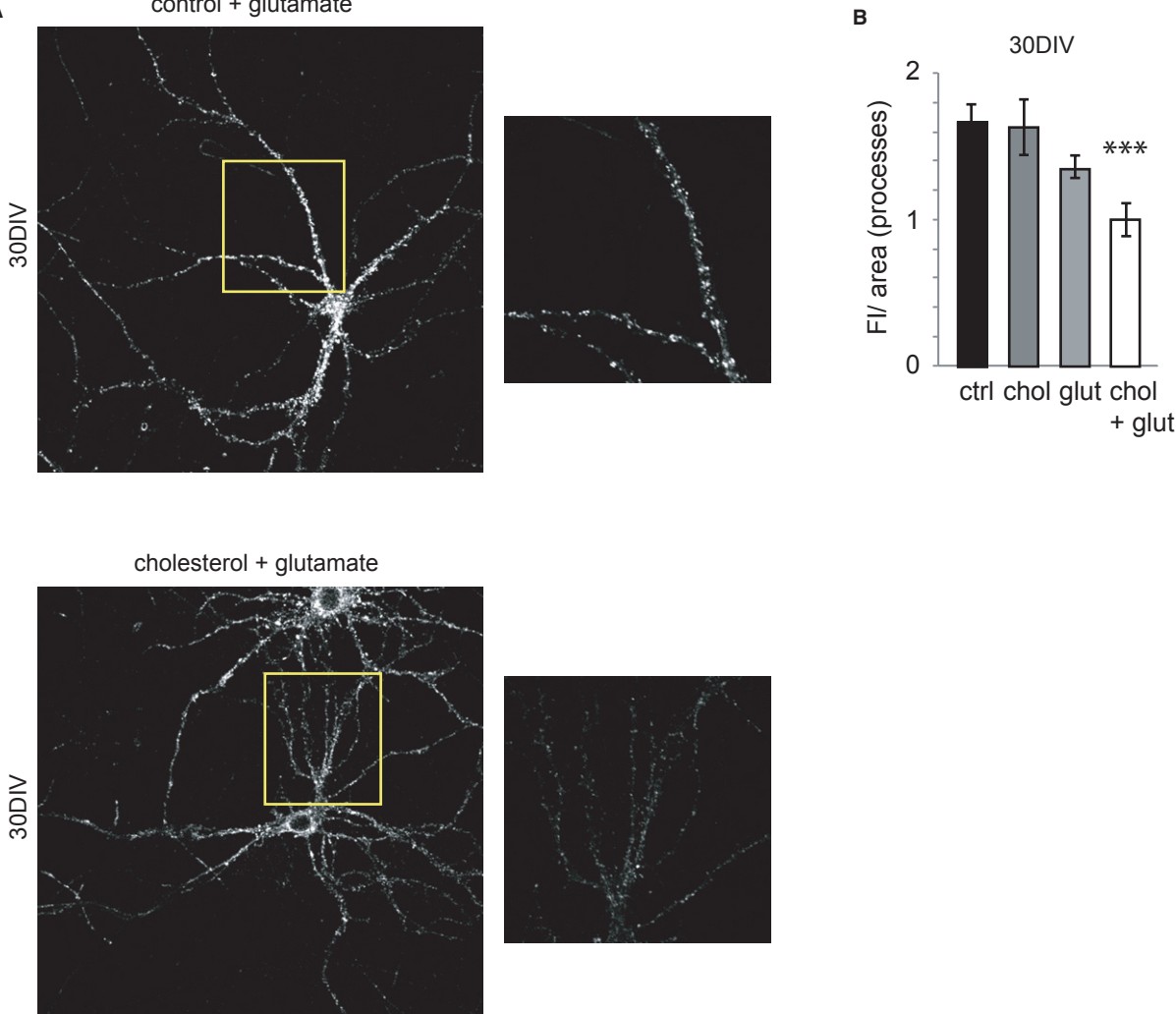

**Figure 4.  Glutamate addition decreases the level of surface AMPARs in cholesterol-replenished 30 DIV neurons.**

A   The surface AMPAR staining decreased in cholesterol-replenished 30 DIV neurons after a 10-min incubation with glutamate. The boxes on the right correspond to high-magnification images of the regions indicated in the insets.

B   The bar plot shows the quantification of the fluorescence intensity (FI)/area (mean $\pm$ s.e.m., relative to the cholesterol + glutamate condition) in the processes of control and cholesterol-replenished neurons. No differences in the FI/area values were found in control, glutamate-stimulated (control = 1.67 $\pm$ 0.12, glutamate = 1.35 $\pm$ 0.07; $P$ = 0.053; $n$ = 3) or cholesterol-replenished unstimulated neurons (control = 1.67 $\pm$ 0.12; cholesterol = 1.63 $\pm$ 0.19; $P$ = 0.87; $n$ = 3). Stimulation of cholesterol-replenished neurons (chol + glut) provoked a significant decrease in the surface GluA2 staining (control = 1.67 $\pm$ 0.12; cholesterol + glutamate = 1.00 $\pm$ 0.12; $P$ = 0.0018; $n$ = 3). The data were compared using Mann-Whitney non-parametric $t$-test.

differences were observed with respect to the D values obtained by glutamate stimulation (Supplementary S4A and B). However, NMDA stimulation of cholesterol-replenished 15 DIV neurons did not produce any differences compared to controls (Supplementary Fig S4B).

**NMDA-LTD is impaired in aged animals in a manner dependent on cholesterol loss**

In light of the fact that the loss of membrane cholesterol in old neurons perturbs AMPAR membrane internalization and lateral diffusion, two key events needed for LTD, this event was further

studied by electrophysiology. Application of NMDA (30 μM) for 4 min induced strong depression in hippocampal slices obtained from 2-month-old (2M) mice, whereas no LTD was observed in slices from 20M mice (Fig 6A). Furthermore, LTD was efficiently induced in slices from middle-aged, 10-month-old (10M) mice, confirming that reduced LTD in the 20M mice is part of the aging phenotype and manifested in early adulthood (Fig 6B).

To determine whether the impaired LTD in the old was due to the loss of cholesterol that occurs naturally in the hippocampus in later life (see Introduction), hippocampal slices from old mice were incubated with the cholesterol replenishment mix (see above) and cholesterol replenishment was seen to rescue NMDA-induced LTD

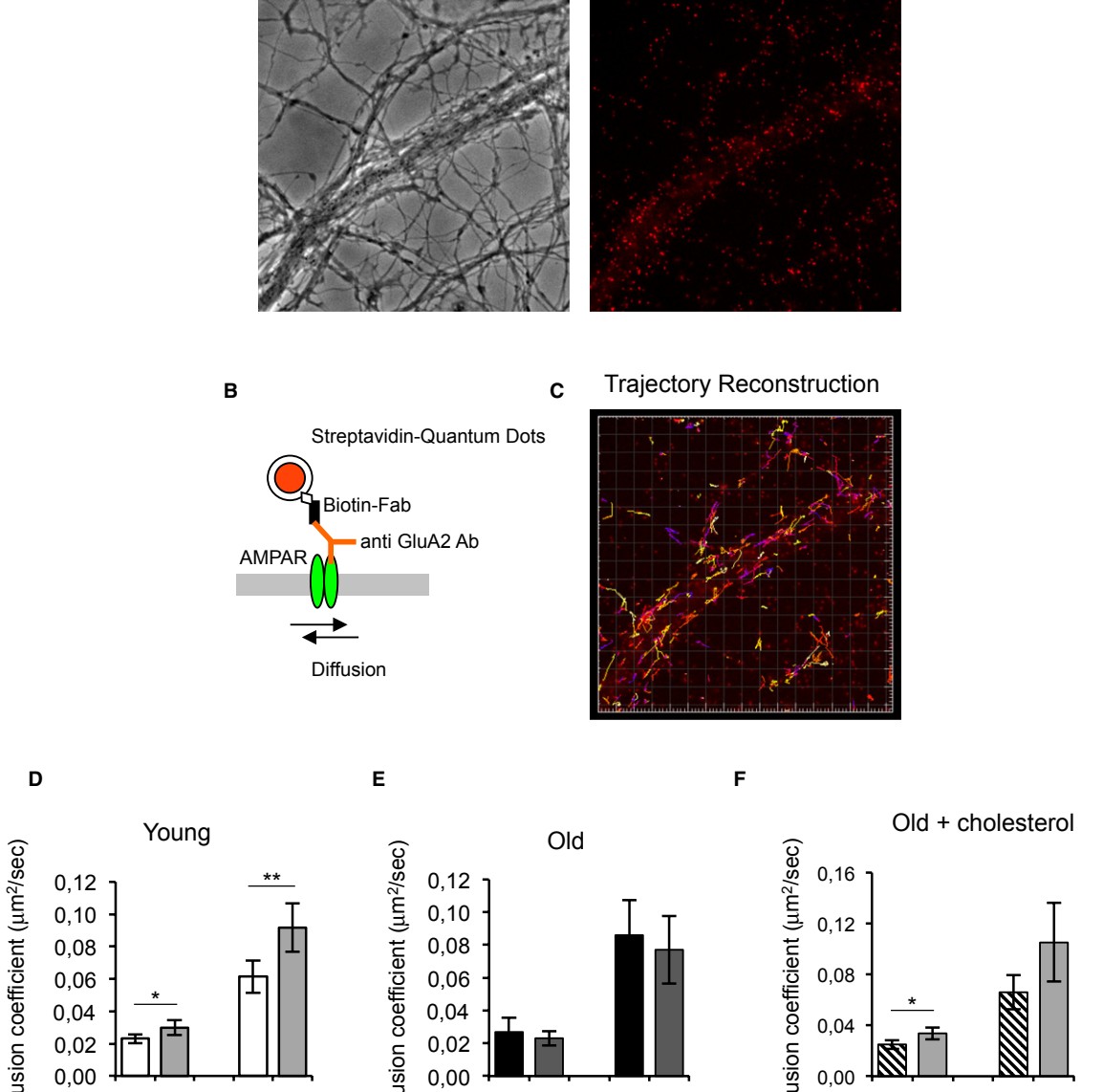

**Figure 5. Lateral diffusion of GluA2-AMPARs is altered in low-cholesterol hippocampal neurons after LTD induction.**

A     Phase contrast and fluorescence images showing the processes of hippocampal neurons where GluA2 subunits were labeled with quantum dots.

B     GluA2-AMPARs were labeled using an anti GluA2 antibody and a biotinylated anti-mouse Fab fragment conjugated to streptavidin-coated quantum dots.

C     Image showing the reconstructed trajectories of individual GluA2-AMPARs.

D, E   Glutamate stimulation increases the diffusion of synaptic and extra-synaptic AMPARs in 15 DIV but not 30 DIV neurons. 15DIV: $D_i$ control = 0.023 $\pm$ 0.0028 µm²/s, $D_i$ glut = 0.0298 $\pm$ 0.0045 µm²/s ($P$ = 0.0201), $D_{out}$ control = 0.0613 $\pm$ 0.0099 µm²/s, $D_{out}$ glut = 0.0917 $\pm$ 0.0149 µm²/s ($P$ = 0.0054). 30 DIV: $D_{in}$ control = 0.0269 $\pm$ 0.0085 µm²/s, $D_{in}$ glut = 0.0231 $\pm$ 0.0044 µm²/s ($P$ = 0.4048) and $D_{out}$ control = 0.086 $\pm$ 0.0213 µm²/s, $D_{out}$ glut = 0.0771 $\pm$ 0.0207 µm²/s ($P$ = 0.5228).

F     The addition of cholesterol to 30 DIV neurons restores the response of synaptic AMPARs to glutamate. $D_{in}$ chol = 0.0250 $\pm$ 0.0035 µm²/s, $D_{in}$ chol + glut = 0.0337 $\pm$ 0.0047 µm²/s ($P$ = 0.0254).

Data information: In individual AMPAR tracking experiments the diffusion coefficients calculated from all the trajectories analyzed (ranging from 250 to 500 obtained from at least five different cultures) have a one-tail distribution. The effect of glutamate in single experiments has been studied using non-parametric Mann-Whitney tests. The mean values of the medians follow a normal distribution and thus groups of experiments were compared using unpaired *t*-test.

in old hippocampal slices (Fig 6A). Strikingly, the LTD in slices from old animals treated with the cholesterol was very similar to that in untreated slices from young mice. To rule-out any unspecific effects, slices from young mice (i.e., with normal cholesterol levels) were exposed to the cholesterol replenishment mix and their electrophysiological response was indistinguishable from that of the controls (Fig 6A). To determine whether cholesterol loss is sufficient to impair LTD, cholesterol was removed from hippocampal slices obtained from 10M mice with the drug MβCD, resulting in a net loss of $28.65 \pm 16.77\%$ cholesterol and a marked reduction in LTD (Fig 6B).

Next, we infused cholesterol for 14 days into the lateral ventricle of 20M mice, after which the animals were sacrificed and hippocampal slices were prepared for electrophysiological recordings. While NMDA induced a brief and transient LTD in the vehicle-treated group, a significant and long-lasting depression was registered in slices from cholesterol-infused aged mice (Fig 6C). No significant differences in the input-output curves were observed in the conditions tested, indicating that basal synaptic transmission is not affected by changes in cholesterol (Supplementary Fig S5).

**Constitutive cholesterol loss impairs LTD in the aged:**
*in vivo* **studies**

To demonstrate the relevance of the age-associated cholesterol loss *in vivo*, we measured LTD in the hippocampus of anesthetized 2M and 20M Wistar rats. This animal model was used as old rats are more resistant than mice to the anesthetic and recording procedures, and as rat aging is also accompanied by a loss of cholesterol in the hippocampus (Martin *et al*, 2008).

To induce hippocampal LTD in rats, concentric bipolar stimulating electrodes were placed in the ipsilateral CA3 field for orthodromic activation of the CA1 field, where evoked potentials were recorded (Fig 7A). In this way, it could be seen that the LTD induced by local application of NMDA was robust in young adult rats and impaired in old animals (Fig 7B).

We tested whether the impaired LTD in old rats could be rescued by enhancing the availability of cholesterol, as observed in mouse hippocampal slices (Fig 6). Before proceeding with the study, we established optimal conditions to guarantee that cholesterol injected into the hippocampus would be able to infuse into a wide enough region to include the area recorded by the electrode. Thus, two injections of a cholesterol solution containing the fluorescent cholesterol

derivative Bodipy-cholesterol (3.5 μM) were applied to the apical dendritic layer of the CA1 region of an adult rat, 1 mm apart (Fig 7A). To verify adequate diffusion of Bodipy-cholesterol to the recording site, hippocampal sections were examined by fluorescence microscopy 60 min later, when the injected Bodipy-cholesterol could be seen to have infused into all the tissue located between the injection sites (Fig 7A). In addition, to avoid tissue damage in the recording zone, the evoked potentials were recorded from an intermediate location between the injection sites (Fig 7A). Having established these parameters, when we examined the electrophysiological recordings, a very strong LTD was evident in old animals that received a hippocampal injection of cholesterol 60 min before LTD induction (Fig 7B). As a control, young rats were also injected with cholesterol 60 min before the LTD induction, although no differences were observed between the treated and untreated animals (Fig 7B). To rule-out variability between animals, the same rat was used as both the experimental and control case, that is, the cholesterol solution was injected into the hippocampus of one hemisphere and control solution into the contralateral hippocampus. The sterol specificity was tested by measuring LTD in aged rats injected in the hippocampus with oleic acid-MβCD or with stigmasterol-MβCD, neither of which restored LTD (Supplementary Fig S6), thereby confirming the specificity of cholesterol in the phenotype observed.

**Chronic central cholesterol treatment enhances spatial learning and memory**

The clear effect of cholesterol replenishment in rescuing impaired LTD in old animals prompted us to test whether the infusion of cholesterol into the brain of old rats would improve some of the cognitive deficits typical of this stage of life. Therefore, 19-month-old rats were perfused in the lateral ventricle for 1 month with a cholesterol replenishment solution (containing the 300 μM cholesterol-MβCD complex + 5 μM free cholesterol dissolved in artificial cerebrospinal fluid – aCSF). Subsequently, the spatial learning ability of the animals was evaluated in the reference version of the Morris water maze, an established hippocampal-dependent task (see Materials and Methods) where aged animals typically exhibit impaired learning of the location of the hidden escape platform (Gallagher *et al*, 1993).

*Post hoc* statistical comparison of the learning curves demonstrated significant differences for the two groups of rats (repeated-measure ANOVA, $F_{1,11} = 5.848$, $P = 0.016$: Fig 8A). A more detailed

**Figure 6.   NMDA-LTD is lost with age and restored by cholesterol replenishment.**

A   NMDA-LTD was observed in slices of 2-month-old mice (2M, ○), but it was absent in 20-month-old mice (20M, ♦), yet cholesterol replenishment restored LTD in old slices (20M + chol, △ gray filled). Cholesterol perfusion did not have any effect on 2M slices (2M + chol, □ gray filled). Empty boxes indicate the period of cholesterol perfusion. Gray squares represent the time of NMDA application. The plot shows the average ± s.e.m. of the responses collected from the last 20 min of the recordings. Values, 2M: $49.05 \pm 2.94\%$; 2M + chol: $30.82 \pm 7.68\%$ ($P^{2M, 2M+chol} = 0.137$); 20M: $82.69 \pm 8.02\%$; 20M + chol: $48.61 \pm 5.50\%$ ($P^2M$, **20M** = 0.014; $P^2$0M, 20M+chol = 0.0093).

B   Cholesterol depletion impaired NMDA-LTD in middle-aged hippocampal slices. Treatment with MβCD 2 h prior to NMDA addition abolished NMDA-LTD in 10-month-old mice. Control (○), cholesterol depleted (■). The average ± s.e.m. of the responses collected from the last 20 min of the recordings for the control ($38.43 \pm 3.68\%$) and MβCD-treated groups ($90.82 \pm 8.45\%$; $P = 0.0003$) are plotted on the right.

C   Intraventricular infusion of cholesterol in old mice restored NMDA-LTD. Field excitatory postsynaptic potentials (fEPSPs) were recorded from hippocampal slices of 24-month-old mice infused with cholesterol (◇) or vehicle (♦) in the lateral ventricle for 14 days. A significant NMDA-LTD was obtained in slices prepared from cholesterol-infused mice, but no LTD was observed in the vehicle-treated group. The plot shows the average ± s.e.m. of the responses collected from the last 20 min of the recordings for control (IV vehicle: $104.02 \pm 3.64\%$) and cholesterol-treated group (IV chol: $68.05 \pm 17.34\%$; $P = 0.0286$).

Data information: The sample traces recorded at the times indicated as 1, 2, and 3 for each condition are shown in (A), (B), and (C). In all the cases, the data were compared using Mann-Whitney non-parametric *t*-tests and the number of animals (*n*) is shown in the figure.

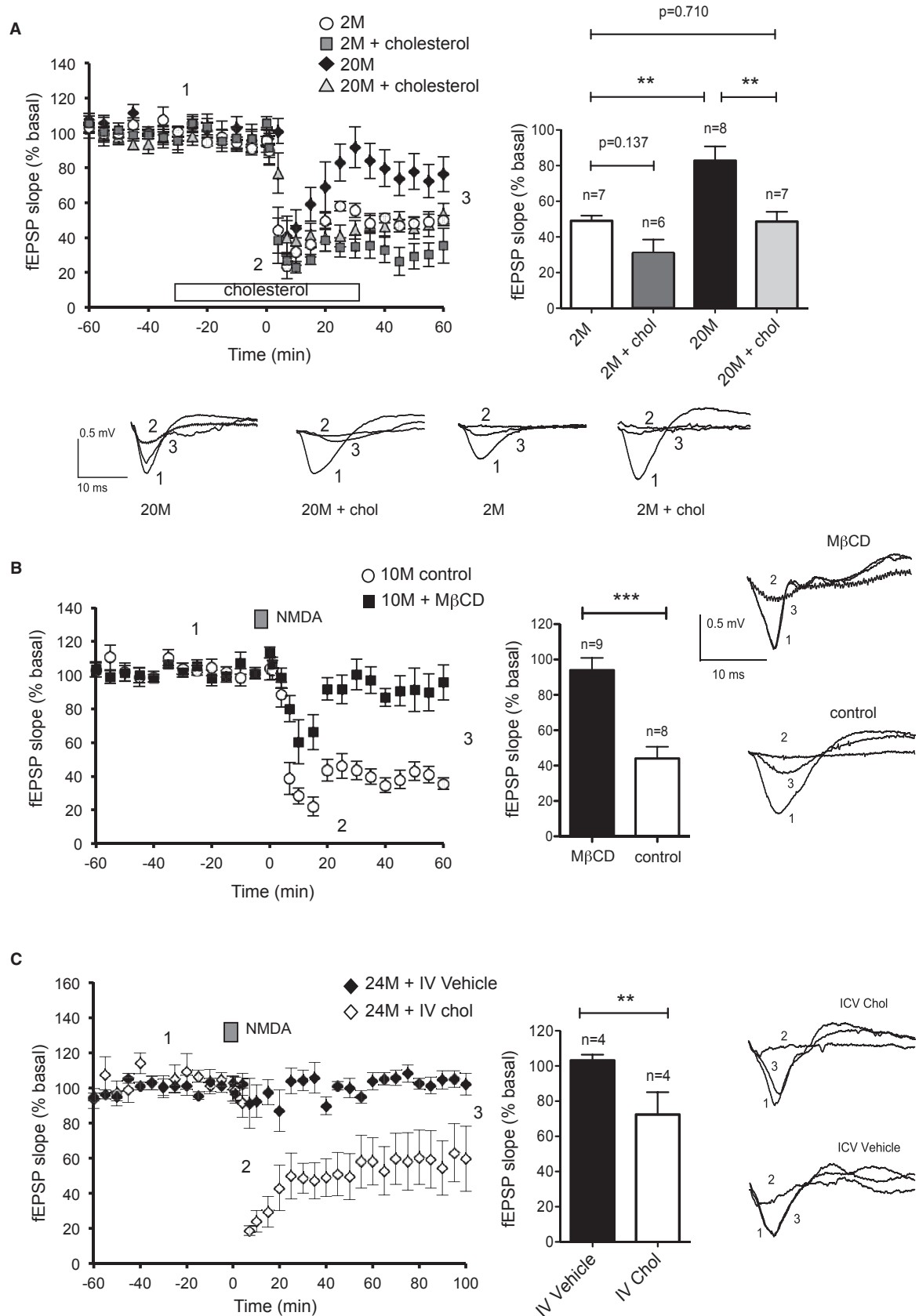

**Figure 6.**

analysis of each training day indicated that cholesterol-treated rats outperformed their controls in the water maze on day 4 ($F_{1,2} = 7,707$; $P = 0.008$). The Bonferroni *post hoc* test indicated significant differences to find the hidden platform between cholesterol-treated rats and controls in trial 11 ($P = 0.019$). One day after the acquisition of spatial learning, the strength of long-term spatial memory was evaluated in a probe trial (i.e., free swimming without platform). During the probe trial, rats that received cholesterol spent more time swimming in the quadrant of the pool that contained the platform during training (target quadrant) than those that did not ($t_{18} = 2.659$, $P = 0.016$: Fig 8B and C). No differences in swim speed or thigmotactic behavior were found during spatial training or in the probe trial. These results show that cholesterol perfusion can rescue learning and memory in old rats.

## Discussion

Hippocampal cholesterol loss during aging occurs in rodents and humans (reviewed in Ledesma *et al*, 2012), and the data presented here suggest that this loss contributes to cognitive decay in the aged. This conclusion is supported by the rescue of the poor LTD in the aged following the addition of cholesterol to hippocampal slices, either in aCSF and directly into the hippocampus. Similarly, intraventricular addition of cholesterol also rescues the reduced learning and memory abilities of old rats.

In addition to the effect of the loss of cholesterol in relation to LTD and cognitive abilities, the data presented here, together with data from previous work, provide evidence for the molecular mechanism underlying this age-dependent impairment of LTD. Indeed, previous studies have shown that the loss of cholesterol in the hippocampus of aged animals drives an increase in TrkB/PI3K activity and p-AKT accumulation (Martin *et al*, 2008, 2011; Trovò *et al*, 2011, 2013). These data are reinforced here by the demonstration that hippocampal extracts from old animals have high levels of active p-Akt. While part of the increased p-Akt activity can be explained by the activity of the TrkB/PI3K pathway (Martin *et al*, 2008), the data presented here indicate that MARCKS detachment may also contribute to the effects of cholesterol loss. In fact, we recently described that MARCKS detachment from hippocampal synapses of old mice is responsible for the decreased PI(4,5)P2 levels and high p-Akt activity (Trovò *et al*, 2013). Thus, in mechanistic terms, the increase in PI3K activity is a major biochemical change determined by the age-associated loss of cholesterol that, together with membrane MARCKS detachment, results in PI(3,4,5)P3 and p-Akt accumulation.

Considering the robust anti-apoptotic role of activated Akt (Airaksinen & Saarma, 2002; Huang & Reichardt, 2003; Zheng & Quirion, 2004), an obvious conclusion from our findings is that the loss of cholesterol with aging plays an important role in the survival of aging neurons. Conversely, the age-associated loss of cholesterol appears to be detrimental for certain functions, and LTD requires NMDA-dependent p-Akt dephosphorylation in order to activate the GSK3β, which in turn regulates AMPAR internalization (Peineau *et al*, 2007). This mechanism does not appear to be operative in old neurons, again most likely due to natural cholesterol loss. Indeed, the levels of p-Akt in response to NMDA are higher in hippocampal extracts from old mice than young mice, and when cholesterol

levels are restored, p-Akt levels diminish significantly to those found in younger animals. Although we cannot rule that the high levels of p-Akt in the aged tissue are the consequence of reduced Akt dephosphorylation by PP1, we have sufficient evidence that part of the problem is the increase in PI3K activity (see Trovò *et al*, 2013; and data herein). Moreover, we show here that AMPARs accumulate at the surface of 30 DIV neurons and that they undergo a very low rate of internalization upon stimulation, both processes that are dependent on membrane cholesterol levels. In fact, PI3K activation has been reported to increase cell surface expression of AMPARs by facilitating receptor insertion (Man *et al*, 2003).

From our data, we infer that AMPAR accumulation at the cell surface is not simply due to changes in membrane composition as a result of cholesterol loss but rather that these receptors are blocked by their interaction with other adaptor molecules. Indeed, single molecule tracking experiments demonstrated that the simple addition of cholesterol at 30 DIV does not augment the $D_{in}$ values in unstimulated conditions. In fact, increases in $D_{in}$ are only observed in cholesterol-treated neurons after glutamate stimulation. This observation concurs with the reduction in surface GluA2-AMPARs only occurring after stimulation in cholesterol-replenished cells. The efficient internalization of AMPARs is essential for LTD, and it requires a decrease in PI(3,4,5)P3 to favor synaptic actin depolymerization, PSD95 degradation, and reduced surface AMPAR expression (Horne & Dell'Acqua, 2007). Accordingly, PSD95 degradation after NMDA stimulation does not occurring in synaptic fractions prepared from old animals or in cholesterol-depleted neurons *in vitro*. Thus, the reduced lateral mobility of synaptic AMPARs after stimulating neurotransmitter can be seen as the consequence of surface accumulation due to defective internalization.

Since MARCKS is required for PIP2 clustering and PLCγ activity (Trovò *et al*, 2013), and therefore for actin depolymerization and receptor internalization, the poor association of MARCKS to the plasma membrane in low-cholesterol membranes can also contribute to the impairment of LTD. The reason for MARCKS detaching from membranes after cholesterol removal remains unclear. One possibility could be that the myristoyl moiety of this protein interacts more weakly with cholesterol-depleted membranes. Furthermore, since high intracellular calcium leads to MARCKS membrane detachment and high calcium is typically a feature of old cells (Arbuzova *et al*, 1998; Trovò *et al*, 2013), the cholesterol loss-dependent detachment of MARCKS could also be driven through an increase in calcium permeability provoked by cholesterol loss.

For practical reasons, our study focused on the consequences and underlying mechanisms of one lipidic change occurring during aging: hippocampal cholesterol reduction. However, it must be born in mind that other lipid alterations also occur at this stage of life, such as the increase in sphingomyelin and ceramide, and the decrease of poly-unsaturated fatty acids (reviewed in Ledesma *et al*, 2012). These changes could also contribute to the cognitive and motor deficits typical at this stage of life, and as a matter of fact, we showed that the SM/cholesterol ratio increases during aging, leading to changes in receptor endocytosis (Trovò *et al*, 2011). Thus, future studies will be needed to determine how other changes in membrane lipid composition during aging participate in the cognitive deficits of the old.

Finally, it is important to note that blood cholesterol does not cross the blood brain barrier, making any beneficial effect of

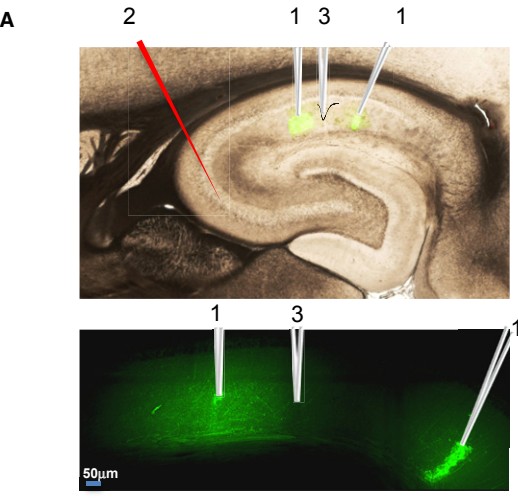

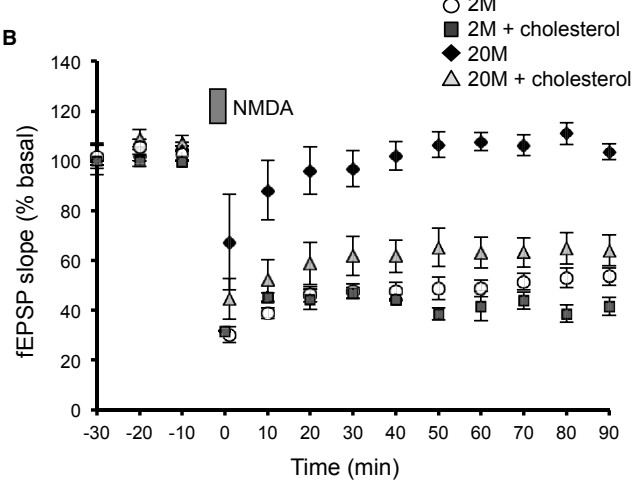

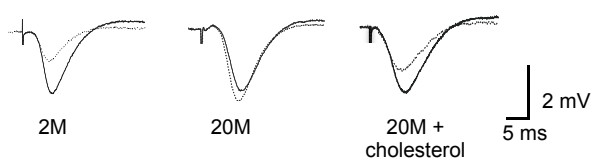

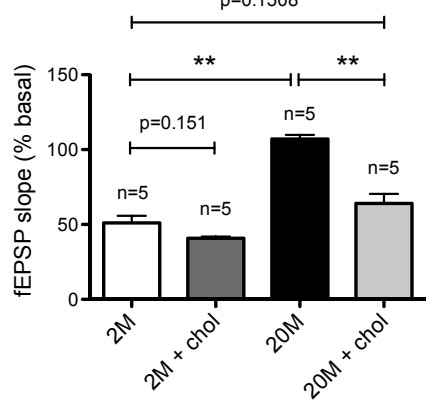

**Figure 7. Cholesterol rescues LTD and cognition in old rats.**

A Sagittal section of the hippocampus of anesthetized rats injected with fluorescent Bodipy-cholesterol. The upper panel shows the merged phase contrast and fluorescence images. The sites where Bodipy-cholesterol was injected (1), the Schaffer stimulation site (2), and the recording site (3) are indicated. Lower panel: magnification of the fluorescence image showing that Bodipy-cholesterol diffuses from the injection sites (1) to the recording zone (2). The image corresponds to the magnification of one section of hippocampus, 60 min after cholesterol injection into the st. radiatum (40× 1.25 IMM OIL Leica laser-scanning confocal microscopy).

B LTD was registered in 2-month-old anesthetized animals (2M, ○), but it was impaired in 20-month-old animals (20M, ♦). Cholesterol injection into the stratum radiatum of the old animals rescued LTD (20M + cholesterol, △ gray filled), but it had no effect in young animals (2M + chol, □ gray filled). The plot shows the evolution of the fEPSPs following the NMDA-inducing LTD protocol ($n = 5$). Average fEPSPs were built over 10-min periods (1 trial per min), and the same individuals were employed for control and cholesterol treatment using the left or right hippocampus. Representative traces are shown of fEPSPs recorded in the stratum radiatum *in vivo* before (solid traces) and 40 min after NMDA-LTD (dotted traces) in 2M, 20M, and 20M rats injected with cholesterol (20M + cholesterol). The plot shows the average ± s.e.m. values of the responses collected from the last 40 min of the recordings from 2M ($51.11 \pm 4.73$), 2M + cholesterol (2M + cholesterol: $40.83 \pm 3.74$; $P^2$M, 2M + chol = 0.151), 20M ($107.03 \pm 2.78$; $P^2$M, 20M = 0.007), and 20M rats injected with cholesterol (20M + chol: $64.12 \pm 6.30$; $P^2$0M, 20M + chol = 0.008). The data were compared using Mann–Whitney non-parametric *t*-tests.

increased cholesterol in the diet unlikely and strongly unrecommended. In fact, the increase in circulatory cholesterol will have a number of deleterious consequences due to the effect in the vasculature (atherosclerosis), consequently affecting the function of all organs. Hence, in theory, strategies to inhibit brain cholesterol loss could improve the cognitive deficits associated with aging.

## Materials and Methods

### Cell cultures and reagents

Hippocampal neurons from 18-day-old rat embryos were cultured for 15 or 30 DIV. Glutamate was added at final concentrations of 100 μM during 15 min, while NMDA was added at 30 μM for 5 min, and the cells were washed and kept in fresh N2 medium for 15 min prior to their analysis. Cholesterol was replenished during 30 min at 37°C using cholesterol-MβCD complex (30 μM, Sigma) supplemented with cholesterol (5 μM) in N2 medium. Cholesterol was depleted by treating 15 DIV hippocampal neurons with MβCD (0.5 mM) for 2 h. Finally, the cholesterol levels obtained after treatments were measured in membrane pellets prepared from hippocampal neurons in culture or from hippocampal slices using the Amplex Red Cholesterol Assay kit (Invitrogen).

### Hippocampal slice recordings

The hippocampus of mice was dissected into ice-cold ACSF that was oxygen-saturated with carbogen (95% $O_2$/5% $CO_2$). Field excitatory postsynaptic potentials (fEPSPs) were recorded using a glass electrode (filled with ACSF, 3–7 MΩ) inserted into the apical dendritic layer of the CA1 area. LTD was induced by applying NMDA (30 μM) dissolved in ACSF for 4 min (Li *et al*, 2004). The cholesterol was added for 60 min as cholesterol-MβCD and 5 μM

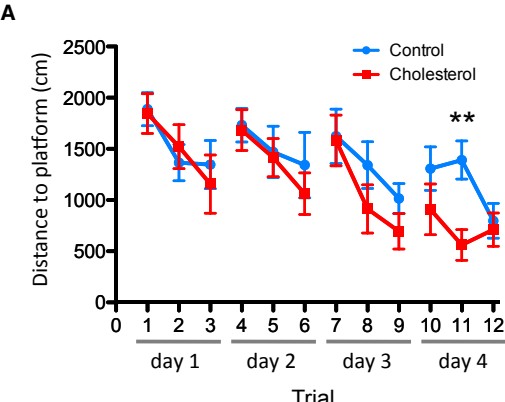

**A**

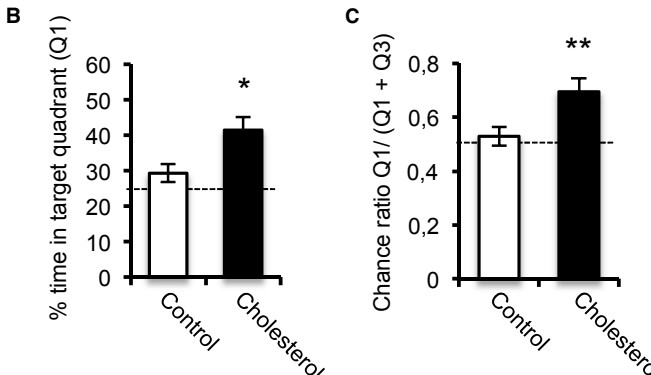

**B**    **C**

**Figure 8.   Cholesterol improves learning and memory in old rats.**

A    The spatial abilities of 20M rats subjected to intra-ventricular infusion of control or cholesterol solution were evaluated in the Morris water maze. The animals were trained three times each day over 4 days, and the learning curves were plotted for the two groups. Average group distance (cm) to find the hidden platform across trials is shown, whereby cholesterol-treated rats ($n = 9$) outperformed their controls ($n = 11$) in the water maze on day 4 (repeated-measure ANOVA, $F_{1,2} = 7,707$; $P = 0.008$), and significant differences were found in trial 11 (Bonferroni *post hoc* test, $P = 0.019$).

B, C    One day after acquisition of spatial learning, the strength of long-term spatial memory was evaluated in a probe trial without the platform. The percentage time spent in the quadrant of the pool that previously contained the platform (Q1) was significantly higher for cholesterol-treated animals compared to the rats that received the vehicle alone (two-sided *t*-test, $P = 0.016$), as shown in (B). Time spent in the target quadrant over the total time spent in the target and the opposite quadrant: Q1/(Q1 + Q3) is shown in (C). While vehicle-treated rats performed at chance level (0.5 represented by a dashed line), cholesterol-treated rats had a good spatial memory index (two-sided *t*-test $P = 0.012$). The results clearly show that cholesterol treatment improved spatial memory in old rats.

cholesterol in aCSF solution. LTD was induced after 30 min in the presence of cholesterol by adding NMDA to the cholesterol solution for 4 min. For cholesterol depletion, hippocampal slices were incubated for 2 h with 0.5 mg/ml MβCD prior to inducing LTD. The drug and control studies were interleaved for both age groups, and the input-output curves were recorded without the addition of picrotoxin.

## Osmotic pump implantation in mice and *in vivo* cholesterol administration

ALZET osmotic mini-pumps (model 2004; ALZET) were filled with either cholesterol solution [300 μM cholesterol-MβCD (Sigma) supplemented with 5 μM cholesterol in aCSF] or the vehicle alone (aCSF), secured by the flow moderator, connected to the brain cannula (Alzet Brain Infusion Kit 3) by flexible tubing, and primed with 0.9% saline for 24 h at 37°C. For implantation, mice were sedated (chloralhydrate 30 mg/ml by intraperitoneal injection) and locally anaesthetized (xylocaine 1% wt/vol, subcutaneous). Subsequently, the cannula was stereotaxically placed into the left lateral ventricle and glued into the skull, while the pump was inserted subcutaneously on the back.

Animals received a continuous infusion of cholesterol or vehicle for 14 days, after which they were sacrificed, their brains removed, and hippocampal slices were prepared for electrophysiological recordings as indicated.

### *In vivo* electrophysiology

Male Wistar rats (2 or 20 months old) were anesthetized with urethane (1.2 g/kg, i.p.) and fastened to a stereotaxic device, maintaining their body temperature at 37°C with a heated blanket. Concentric bipolar stimulating electrodes were placed in the ipsilateral CA3 field (mm from bregma, AP: 1.2; L: 2.6; V: 3.5) for orthodromic activation of the CA1 field. The stimulus intensity was set to produce a CA1 population spike of ~40% of the maximum amplitude. Recording and drug injections were achieved through glass micropipettes (1–4 MΩ) connected to a high-impedance headstage. The signals were amplified and acquired (10 kHz sampling rate) using low noise multi-channel system recording hardware and software (Reutlingen, Germany).

Cholesterol was pressure-ejected into the apical dendritic layer of CA1 through a recording pipette filled with cholesterol-MβCD complex (300 μM) plus cholesterol (5 μM, Sigma-Aldrich). Two injections delivered 1 mm apart were applied 60 min before LTD was induced, while subsequent evoked potentials were recorded from an intermediate location. Averaged evoked fEPSPs were collected each minute before (30 min) and for up to 90 min after LTD induction. LTD was induced in st. radiatum of CA1 region by local application of NMDA (NMDA 500 μM plus NaCl 1 M). The volume was adjusted to avoid noticeable effects on evoked potentials in control experiments when only solvent was injected (NaCl 1 M). The fEPSP was measured as the maximum slope of the initial negative phase.

### Bodipy-cholesterol injection and histology

Fluorescent Bodipy-cholesterol (5 mM stock in DMSO, Invitrogen) was dissolved in saline solution (at 3.5 μM, maximum solubility) and injected into the st. radiatum of the CA1 area (apical dendritic layer). Two glass pipettes were placed 500 μm anteroposterior to the recording pipette, and the microdrops (two drops on both sides, 1 mm apart) were adjusted (~200 nl) to limit the volume of the bathed tissue. The LTD protocol commenced 60 min later. Finally, the brains were removed and 100 μm sagittal cryostat sections were rapidly obtained. Slices were examined by fluorescence microscopy

(560–610 nm) to determine the extension of the tissue infused with cholesterol.

### Immunocytochemistry

To identify surface AMPARs, unpermeabilized control or treated neurons were fixed and stained with an antibody recognizing an extracellular epitope of the rat GluA2 subunit (BD pharmingen). To measure the ratio of surface/total AMPARs, antibody-feeding experiments were performed. Briefly, live cells were incubated with the same anti GluA2 antibody (5 μg/ml) for 10 min at 20°C in conditioned cell medium to minimize endocytosis. The cells were then washed briefly and returned to the medium at 37°C for agonist treatment and staining, as indicated elsewhere (Carroll *et al*, 1999). Surface and internalized AMPARs were labeled with fluorescent secondary antibodies (Invitrogen) before and after permeabilization respectively. The fluorescence intensity corresponding to the surface and internalized GluA2 was measured in a defined area, and the ratio of the internalized GluA2 was calculated as internalized GluA2/(internalized + surface GluA2). The values were corrected for the control (unstimulated) conditions, considered as 100%. For each experiment, the same parameters were used to acquire the images in all the conditions.

Internalized GluA2-AMPARs were measured using the same antibody-feeding protocol followed by acid stripping. In these experiments, the antibodies bound to receptors on the cell surface after agonist treatment was removed. The cells were chilled in ice-cold TBS and washed with 0.5 M NaCl/0.2 M acetic acid for 4 min (Carroll *et al*, 1999). The intracellular AMPARs were identified by immunofluorescence following cell permeabilization.

### GluA2 live cell staining and single-particle tracking

GluA2-containing AMPARs were labeled with quantum dot nanoparticles (Invitrogen), and single-particle tracking and their trajectory were analyzed as described in detail in the Supporting Information.

### Rat surgery and osmotic pump implantation

A chronic brain infusion cannula (Alzet brain infusion kit 2-Alzet Corporation, Palo Alto, CA, USA) was implanted into the right lateral ventricle of the brain by stereotaxis at the following coordinates relative to bregma: −0.85 mm posterior, + 1.1 mm lateral, and −4.0 mm deep according to the atlas of Paxinos and Watson (Paxinos *et al*, 1985). The infusion cannula was attached through catheter tubing to an Alzet miniosmotic pump (model 2006; flow rate: 0.15 μl/h). Approximately 36 h prior to subcutaneous implantation of the osmotic minipump into the rat's midscapular region, the osmotic minipump was primed by immersion in sterile 0.9% saline at 37°C.

To determine whether the beneficial effects observed at the electrophysiological level have a cognitive counterpart, 19-month-old rats were intraventricularly perfused for 4 weeks with a solution of cholesterol-MβCD (300 μM) + cholesterol (5 μM) dissolved in aCSF. Subsequently, the spatial ability of the animals was evaluated in the reference version of the Morris water maze.

### Water maze learning – Spatial training

The water maze (Morris, 1984) was a black circular pool (2 m diameter, 45 cm high) filled with water (30 cm deep) at 24 ± 1°C. An invisible escape platform (11 cm diameter) was placed at a fixed location equidistant from the sidewall and middle of the pool and submerged 1.5 cm below the surface of the water. The behavior of the animal (latency, distance, swim speed, and navigation path) was monitored using a video camera mounted on the ceiling above the center of the pool and a computerized tracking system (Ethovision 1.90, Noldus IT, the Netherlands). Rats completed three trials per day over four consecutive days using a 60 s intertrial interval and a 90 s cut-off to locate the platform in a trial. If the animal failed to escape within 90 s, it was guided to the platform by the experimenter and it was allowed to stay there for 30 s. During the training procedure, the location of the platform remained constant in one virtual quadrant of the maze, while the starting position for each trial varied among the four equally spaced positions around the perimeter of the maze. On day 5, we conducted a probe trial in which the escape platform was removed from the pool and the rat was allowed to swim for 60 s. The trial began with the rat in the quadrant opposite to that which previously housed the training platform and the time spent in each quadrant was recorded. In order to discriminate against the level of chance, a ratio was defined as the percent time spent in the target quadrant over the total time spent in the target and the opposite quadrant (chance level at 50%). In addition, the time the animal needed to cross the exact point of the location of the platform for the first time during the probe trial was used as an indicator of spatial memory. Cognitive evaluation of the spatial abilities of the rats was performed by a trained experimenter blind to the animals' treatment.

### Statistical analysis

Groups of data with a normal distribution and equal variances were compared using the unpaired parametric *t*-test, while data with different variances were compared using the unpaired *t*-test with Welch's correction. Groups of data where a Gaussian distribution could not be assumed were compared with non-parametric Mann-Whitney test.

### Animal handling

All the experiments were performed in accordance with European Union guidelines (2010/63/UE) regarding the use of laboratory animals. Thirteen aged (20–24 months) and five young adult (2 months) Male Wistar rats were used for *in vivo* LTD, and 24 old male Wistar rats were used for the water maze experiments. Seventeen old (20 months) and 17 adult (4 and 10 months) male C57BL6 mice were used for LTD measurements, and approximately 30 animals of each age were used for biochemical experiments performed in acute hippocampal slices. Embryos from pregnant E18 Wistar rats were used to prepare hippocampal primary cultures.

**Supplementary information** for this article is available online: http://embomolmed.embopress.org

## The paper explained

### Problem

Cognitive decline is one of the many characteristics of aging as most individuals older than 65 develop cognitive deficits or age-associated memory impairments. Accordingly, it is noteworthy that the hippocampus, a structure mainly involved in learning and memory, is particularly sensitive to aging. However, these impairments are not the result of massive cell death, indicating that more subtle mechanisms must be affected by aging to produce memory decline.

### Results

Our manuscript addresses the relationship between the low membrane cholesterol levels found in the hippocampus of old rodents with the reduced cognition that characterizes this stage of life. We demonstrate that restoring cholesterol levels in the aged hippocampus to the values found in young adults can rescue learning and memory in the old. Furthermore, we explain the mechanisms involved in memory loss during aging.

### Impact

Although no-one in the field of synaptic plasticity, or in the field of brain dysfunction with aging, would deny the importance of membrane lipids, particularly of cholesterol, for either of these processes, relevant information is scarce if not completely lacking. In fact, nobody has ever addressed relationship between cholesterol loss and cognition deficits in the old. For this reason, we think that our conclusions will be of interest to those interested in both synaptic plasticity and aging, with also a high translational potential to develop drugs to treat age-related cognition impairment.

## Acknowledgements

We thank Kristel Vennekens and Irene Palomares for their technical assistance, and Lola Ledesma, Jose A. Esteban and Shira Knafo for critical reading of the manuscript and suggestions. This work was supported by the Belgian Flanders Fund for Scientific Research (FWO G 0.666.10N), NEUROBRAINNET IAP 7/16, the Flemish Government Methusalem Grant and the Spanish Ministry of Science (SAF2010-14906), and Innovation Ingenio-Consolider (CSD2010-00045) grants, all to CGD. This work was also supported by the FWO Grant G0D7614N to DB and CGD. The following institutions provided financial support to CGD: Fund for Scientific Research Flanders (FWO), the Federal Office for Scientific Affairs (IUAP P6/43), Flemish Government's Methusalem Grant, NEUROBRAINNET grant from the Belgian Government (IAP 7/16) and Spanish Ministry of Science, Innovation Ingenio-Consolider CSD2010-00064 and Plan Nacional SAF2010-14906. SAM is a Marie Curie fellow, FP7-PIIF-GA-2009-252375.

## Author contributions

MGM contributed to the conception, design, analysis and interpretation of the experiments, the data acquisition for AMPAR dynamics and lateral diffusion, the behavioral tests, p-AKT accumulation and for drafting the article. CGD contributed to the conception, design, analysis and interpretation of the experiments, and the drafting of the article. IS contributed to the acquisition, analysis and interpretation of the data on PTEN and PSD95. AK and OH contributed to the conception, design, analysis, and interpretation of the LTD recordings *in vivo*. TA and DB contributed to the design, analysis, and interpretation of the LTD recordings in hippocampal slices. CV contributed to the design, analysis, and interpretation of the behavioral tests. SAM contributed to the analysis and interpretation of the data on AMPAR lateral diffusion. SM contributed to the design of the experiments into AMPAR lateral diffusion.

## Conflict of interest

The authors declare that they have no conflict of interest.

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
