## [Review Process File · EMBO Molecular Medicine]

Constitutive Hippocampal Cholesterol Loss Underlies Poor Cognition in Old Rodents

Mauricio G. Martin, Tariq Ahmed, Alejandra Korovaichuk, Cesar Venero, Silvia A. Menchón, Isabel Salas, Sebastian Munck, Oscar Herreras, Detlef Balschun and Carlos G. Dotti

Corresponding authors: Carlos Dotti and Mauricio Martin, CSIC/UAM and VIB/KU Leuven

Review timeline:

Submission date:	22 November 2013
Editorial Decision:	16 January 2014
Revision received:	19 March 2014
Editorial Decision:	11 April 2014
Additional Author Correspondence:	11 April 2014
Additional Editorial Correspondence :	14 April 2014
Accepted:	29 April 2014

Transaction Report:

Editor: Céline Carret

1st Editorial Decision

16 January 2014

Thank you for the submission of your manuscript to EMBO Molecular Medicine and please accept my sincere apologies for the delay due to a combination of holiday season and a late referee. We have now heard back from the three referees whom we asked to evaluate your manuscript. Although the referees find the study to be interesting, they also raise a couple of concerns that need to be addressed in the next version of your manuscript.

As you will see from the comments below, all three referees appreciate the biological insight and clear general interest of the data. Nevertheless, referees 1 and 2 suggest to add an important missing control (cholesterol treatment of young mice); referee 2 regrets the limited mechanistic insights provided and suggests a certain number of experiments to improve the conclusiveness of the study; finally referees 2 and 3 also request clarifications and better explanations. In addition, and as suggested by both referees 1 and 2, it would be nice if you could thoroughly proofread your article for grammar and spelling.

Overall, we would be happy to consider a revision of your manuscript if you can address the issues that have been raised experimentally when appropriate. Please note that it is EMBO Molecular Medicine policy to allow only a single round of revision and that, as acceptance or rejection of the manuscript will depend on another round of review, your responses should be as complete as possible.

I look forward to seeing a revised form of your manuscript.

***** Reviewer's comments *****

Referee #1 (Remarks):

This is a very interesting and potentially actionable study, and I recommend publication in EMBO after minor revisions.

Please give some more information about why cholesterol should affect Akt phosphorylation. Are lipid rafts involved? Is this completely unknown?

The effects of cholesterol are striking and might have general benefit for the elderly. Can they be produced just by adding cholesterol to the food?

The authors investigate the effect of cholesterol on old rats, but there are no young-rat controls. Perhaps cholesterol just has a general beneficial effect on learning independent of age. This control (actually an experiment) needs to be included; not necessarily in all the studies, but in several at least.

Finally, and quite important, the paper needs to be written in proper English. Please find a good editor and fix the grammar and spelling.

Perhaps it's just a grammatical problem, but on Page 4 (and later in the text), the authors state that they are setting out to "prove" their hypothesis. This implies a biased approach. They should say "to test this prediction" rather than, for example "to prove this prediction". Here is their sentence: "To prove this prediction, p-Akt levels were measured in acute hippocampal slices of young (4 month old) and old (20 month old) mice, in un-stimulated and after pharmacological LTD induction."

Referee #2 (Remarks):

This is an interesting study addressing the role of age-dependent cholesterol decline in neuronal function. Authors show that phosphorylated Akt accumulates in hippocampal slice cultures of aged mice. Levels of pAkt were sensitive to NMDA treatment of slices from young, but not from older mice. pAkt levels in slices from aged mice were decreased by addition of cholesterol. Authors also investigated the effects on GluR expression and mobility at the cell surface and its internalization in primary neuronal cultures at different times in vitro.

In another set of experiments authors investigated the effect of cholesterol supplementation on LTD in slice cultures and learning and memory in living mice and rats. In these experiments cholesterol application partially reversed the age-dependent deficit in older animals.

The spectrum of applied methods is impressive ranging from biochemical and cell biological approaches to electrophysiology and cognition tests in live animals.

However, the manuscript also reveals a number of caveats on the functional connection of the different findings. In particular, the involvement of the proposed molecular mechanisms in the different observations needs to be strengthened. It should also be tested if cholesterol application exerts beneficial effects in young animals.

Specific comments:

1. The blot in fig. 1a indicates that NMDA increases total Akt level, but decreases pAkt (in 4 month slices). However, only ratios of pAkt/total Akt are provided. Authors should also provide quantitative data on total Akt levels (normalized to actin) and pAkt levels separately to assess whether NMDA affects expression of Akt or selectively its phosphorylation. It is stated that NMDA "triggered a dramatic dephosphorylation of Akt..." (p.4, description of fig. 1a). The experiment does not allow a differentiation between inhibition of phosphorylation or stimulation of dephosphorylation. Specific kinase and phosphatase modulators could be used to assess both possibilities.
2. Authors only show the effect of cholesterol on 20 M slices (fig. 1b). It should be also tested whether cholesterol affects Akt expression and phosphorylation in 4 M slices. This is important to evaluate whether cholesterol has a negative effect on Akt phosphorylation also at younger age or selectively in the old. Similarly, the effect of cholesterol addition on MARCKS distribution should be shown for young mice (fig. 1c). The findings on MARCKS association with membranes are not well connected to the other data in this figure and appear descriptive. How does cholesterol affect membrane localization of MARCKS? To demonstrate a functional connection of MARCKS to phosphorylation of Akt, the effects of cholesterol extraction or addition on pAkt should be tested upon down-regulation or overexpression of MARCKS. Levels of MARCKS in the soluble fraction should also be analyzed. Why only cells at 15 DIV were analyzed? Later in the study authors show 15 DIV ('young') and 30 DIV (old) neurons.
3. Levels of GluR2 at the cell surface were analyzed by stainings with antibodies and fluorescence quantification. According to the information in the methods section cells were incubated 20 min at 20°C. However, under these conditions antibodies/receptors could also be internalized. So it appears unclear whether only cell surface GluR2 is visualized with this method. Studies should also be complemented by specific labeling of surface proteins with biotin and biochemical detection of GluR2 by immunoblotting after precipitation with streptavidin.
4. For the supplementation with cholesterol only 30 DIV neurons were used (fig. 4). Cholesterol addition alone did not affect surface expression of GluR2? This requires explanation. When cholesterol loss would be responsible for the increased surface levels of GluR2 in older neurons (fig. 3), then why cholesterol supplementation in old neurons does not decrease GluR2 levels at the surface? In fig. 3, authors claim different surface expression of GluR2 in young and old neurons without glutamate stimulation. It is unclear what n=3 means (figure legend); 3 experiments, 3 cells, 3 areas? More information on the quantification should be provided. Does cholesterol with or without glutamate stimulation affect GluR2 internalization in 15 DIV neurons?
5. 15 DIV cells should also be included in the experiments on cholesterol effects on lateral mobility of GluR2 (fig. 5).
6. For LTD measurements, cholesterol has been applied 30 min before to 30 min after NMDA stimulation. To test whether cholesterol exerts sustained effects of LTD, slices should be pre-treated with cholesterol before NMDA stimulation and then analyzed in absence of cholesterol. In turn, cholesterol should only be added after or together with NMDA stimulation. These variations might allow distinguish between sustained and acute effects of cholesterol on LTD.
7. Effects of cholesterol extraction by MCD on LTD are only shown for 10 M mice (fig. 6b). Does MCD treatment affect LTD in younger (2M) mice? Similar question arises for the experiments with rats (fig. 7 and 8). Does cholesterol application affect LTD and memory in younger animals? This appears important to assess whether cholesterol application generally strengthens LTD or selectively reverts deficits in aged mice with gradual loss of cholesterol.

Additional points:

- the error bars should also be provided for controls in fig. 1.
- information on the number of experiments and samples, cells should be provided for each

quantification (fig. 1a,b,c; fig.2; fig.3c, fig.5;).
-reference 'Jurado et al., 2010' cited on p.6 is missing in the reference list
- the ms should be carefully checked for phasing and grammatical errors.

Referee #3 (Comments on Novelty/Model System):

This is a very interesting study implicating lowered cholesterol content in the impairment of LTD during aging. For the most part the experiments are well designed and the results convincing. I do have, however, some reservation about the relevance of the studies concerning the internalization and mobility of the AMPA receptors. Those were done in cell cultures where LTD was induced with glutamate and monitored by imaging GluR2; in contrast, the evaluation of LTD in the aged tissue was done in slices, using NMDA and monitored as changes in synaptic response. First, the loss of internalization observed in the 30 days in the life of a culture cell it is not a model for the loss of LTD that occurs in the course on months. Second, glutamate (as used in the culture setting) can induce two different form of LTD: NMDAR-dependent LTD, which is not dependent on GLuR2, and mGluR-dependent LTD, which does depend on GluR2. It must be noted, however, that even without those experiments, the case for a crucial role of lowered cholesterol in the impairment of LTD during aging is still strong. Perhaps a simple solution will be to remove figures 3 and 4.

The citations reporting a loss of LTD during aging need some working. There are changes of LTD throughout the lifespan and at different times. Some are maturational processes that occur within the few postnatal weeks (Dudek and Bear, Bear and Abraham, Milner, Kemp), other occur during middle age or after (Amhed et al, Lee et al)

Other points

The reference Jurado et al missing

Most of the acronyms are introduced without definition, making it difficult for a non-specialist. For example p-Akt was brought to the introduction without any warning.

Referee #3 (Remarks):

It will be a very nice study if cell culture experiments are removed or better justified

Point-by-point answer to reviewers' queries.

Referee #1.

Comment: Please give some more information about why cholesterol should affect Akt phosphorylation. Are lipid rafts involved? Is this completely unknown?

Answer: we have made the corresponding clarification in the text (underlined, page 3).

In an early publication (Martin et al., 2008, Cholesterol loss enhances TrkB signaling in hippocampal neurons aging in vitro. Mol. Biol. Cell 19(5); 2101-2112), we showed that cholesterol loss during aging induces the ligand independent clustering and activation of the receptor tyrosine kinase TrkB, leading to the increase in PI3K activity and phosphorylation of Akt. These events occur in cholesterol-deficient detergent-resistant membrane domains. In a subsequent publication (Trovò, L. et al 2011, Sphingomyelin upregulation in mature neurons contributes to TrkB activity by Rac1 endocytosis. J. Cell Science 124(8): 1308-15), we reported that the increase in TrkB activity in cholesterol poor membranes was due to the increase of the ratio sphingomyelin/cholesterol, indicative that cholesterol loss during aging leads to the formation of a different type of signaling platforms (rafts). Thirdly, in a recent publication (Trovò, L., et al 2013, Low hippocampal PI(4,5)P2 contributes to reduced cognition in old mice due to loss of MARCKS. Nature Neuroscience 16, 449-455) we identified that increased PI3K/Akt activity in old neurons may also result from increased IL-1B activity because IL-1 signaling is enhanced in lipid rafts.

Comment: the effects of cholesterol are striking and might have general benefit for the elderly. Can they be produced just by adding cholesterol to the food?

Answer: cholesterol does not cross the blood brain barrier, making any beneficial effect of increased cholesterol in the diet unlikely. In addition, the increase in circulatory cholesterol will have a number of deleterious consequences, due to the effect in the vasculature (atherosclerosis) and, as consequence, in all organs' function.

On the other hand, for example phytosterols and phytosterol esters differ from cholesterol by their ability to cross the blood brain barrier and replace cholesterol in cellular membranes. Hence, in theory, increasing the intake of those sterols or inhibiting cholesterol degradation in the brain, could improve the cognition deficits of the old. These are subject of our current work. A paragraph commenting this has also been included in the discussion.

Comment: The authors investigate the effect of cholesterol on old rats, but there are no young-rat controls. Perhaps cholesterol just has a general beneficial effect on learning independent of age. This control (actually an experiment) needs to be included; not necessarily in all the studies, but in several at least.

Answer: we now present four new figure panels showing that cholesterol addition to young neurons does not result in any beneficial effect on cognition-associated mechanisms. In the New Figure 1C and on page 3 (underlined) we show that cholesterol addition does not

potentiate Akt activity in hippocampal slices prepared from young mice. The Supporting information Figure 4B shows that cholesterol addition does not improve AMPA receptor lateral diffusion in 15DIV cultured hippocampal neurons after NMDA stimulation. The new Figure 6 (underlined on page 11) shows that cholesterol replacement does not improve LTD in slices from young mice, and the new Figure 7B (underlined in page 12) depicts that hippocampal cholesterol perfusion does not improve LTD in young rats *in vivo*.

Comment: Finally, and quite important, the paper needs to be written in proper English. Please find a good editor and fix the grammar and spelling. on Page 4 (and later in the text), the authors state that they are setting out to "prove" their hypothesis. This implies a biased approach. They should say "to test this prediction" rather than, for example "to prove this prediction".

Answer: the paper was now revised and re-edited by an English-speaking (native) colleague. We have also made the other correction (to test instead of to prove).

Referee #2.

Comment 1. The blot in fig.1a indicates that NMDA increases total Akt level, but decreases pAkt (in 4month slices). However, only ratios of pAkt/total Akt are provided. Authors should also provide quantitative data on total Akt levels (normalized to actin) and pAkt levels separately to assess whether NMDA affects expression of Akt or selectively its phosphorylation. It is stated that NMDA "triggered a dramatic dephosphorylation of Akt..." (p.4, description of fig.1a). The experiment does not allow a differentiation between inhibition of phosphorylation or stimulation of dephosphorylation. Specific kinase and phosphatase modulators could be used to assess both possibilities.

Answer: The normalization of Akt to actin is now provided, showing that Akt levels does not change, and the amount of p-Akt decrease only in 4M mice after NMDA stimulation (see panel A in Figure 1). Regarding the second point: we made that statement based on numerous previous publications where it was demonstrated that NMDA induces Akt dephosphorylation through the activation of the specific phosphatase PP1. This is now clarified in the text (see underlined, page 4).

Comment 2. Authors only show the effect of cholesterol on 20 M slices (fig.1b). It should be also tested whether cholesterol affects Akt expression and phosphorylation in 4 M slices. This is important to evaluate whether cholesterol has a negative effect on Akt phosphorylation also at younger age or selectively in the old. Similarly, the effect of cholesterol addition on MARCKS distribution should be shown for young mice (fig. 1c).

The findings on MARCKS association with membranes are not well connected to the other data in this figure and appear descriptive. How does cholesterol affect membrane localization of MARCKS ? To demonstrate a functional connection of MARCKS to phosphorylation of Akt, the effects of cholesterol extraction or addition on pAkt should be tested upon down-regulation or overexpression of MARCKS. Levels of MARCKS in the soluble fraction should also be analyzed. Why only cells at 15 DIV were analyzed ? Later in the study authors show 15 DIV ('young') and 30 DIV (old) neurons.

Answer

We now show the effect of cholesterol addition on Akt activity in young animals. The data are included in the new Figure 1C. From these series of new data it is possible to conclude that the beneficial effect of cholesterol addition on Akt activity is selective for the old. We also included the Fig. 1E where we show that cholesterol addition to acute hippocampal slices from young mice does not have effect on membrane MARCKS distribution (also underlined text on page 7).

We also show now that MARCKS knock down (by treatment of neuronal cultures with lentiviral particles expressing the shRNA) enhances the phosphorylation state of Akt. We also show that this p-Akt increase triggered by MARCKS knock down is further enhanced by cholesterol removal, thus establishing a direct link between these two events. These results are included in the new Supporting Figure 1 (underlined on page 7)

The levels of MARCKS in the soluble fraction of cholesterol-depleted cells were analyzed. The western blot is included in figure 1D and text is underlined on page 6.

In a previous work developed in our laboratory (Trovò, L., et al 2013, Low hippocampal PI(4,5)P2 contributes to reduced cognition in old mice due to loss of MARCKS. Nature Neuroscience 16, 449-455) it was shown that MARCKS detaches from the membrane as hippocampal neurons age *in vivo*.

In this paper we show that cholesterol loss is the upstream event triggering that detachment. We used 15DIV neurons treated with cyclodextrin simply as a tool to demonstrate that in fact, the cholesterol loss is the specific event involved in MARCKS detachment from plasma membrane, and no other process that could occur *in vitro* between 15 and 30DIV.

Comment 3. Levels of GluR2 at the cell surface were analyzed by stainings with antibodies and fluorescence quantification. According to the information in the methods section cells were incubated 20 min at 20°C. However, under these conditions antibodies/receptors could also be internalized. So it appears unclear whether only cell surface GluR2 is visualized with this method. Studies should also be complemented by specific labeling of surface proteins with biotin and biochemical detection of GluR2 by immunoblotting after precipitation with streptavidin.

Answer: we are afraid we did not explain this experiment in a proper manner. This is now clarified in the text (underlined on page 21-22).

We performed three different protocols to visualize GluA2-AMPA receptors:

- i) To identify GluA2 at the cell surface, the immunolabeling of fixed, non-permeabilized neurons was performed (Figure 3A and Figure 4).
- ii) The pre-incubation at 20°C during 10 min was performed to allow antibody-receptor complex formation minimizing endocytosis. Then, the cells were treated with agonists at 37°C to allow endocytosis to occur. After the treatment, the antibodies bound to receptors at the cell surface were washed out (acid stripping). The cells were fixed, permeabilized and stained with the secondary antibody. Only internalized receptors were detected by this method (Figure 3D and 3E).
- iii) To measure the ratio internalized / total receptors, the same protocol as in ii) was performed but in this case cells were fixed without acid stripping. Then both, surface and internalized receptors were labeled by incubation with two different secondary antibodies before and after permeabilization respectively (Supporting Information Fig. 2B).

Comment 4. For the supplementation with cholesterol only 30 DIV neurons were used (fig.4). Cholesterol addition alone did not affect surface expression of GluR2? This requires explanation. When cholesterol loss would be responsible for the increased surface levels of GluR2 in older neurons (fig.3), then why cholesterol supplementation in old neurons does not decrease GluR2 levels at the surface? In fig. 3, authors claim different surface expression of GluR2 in young and old neurons without glutamate stimulation. It is unclear what n=3 means (figure legend); 3 experiments, 3 cells, 3 areas? More information on the

quantification should be provided. Does cholesterol with or without glutamate stimulation affect GluR2 internalization in 15 DIV neurons?

Answer: our interpretation to this query is the following: glutamate addition triggers receptor internalization through the activation of a number of events, e.g. phosphatases activation, actin de-polymerization and removal of several adaptor and scaffolding proteins. Our results indicate that when cholesterol is low in the membrane, these events cannot occur. In fact, we show that cholesterol loss results in p-Akt accumulation. Thus, during LTD, inefficient Akt dephosphorylation impairs GSK3 β activation, required for receptor internalization. These events are rescued by the addition of cholesterol: p-Akt levels are reduced to basal and receptor internalization after stimulation is more efficient. In control (glutamate unstimulated), 30DIV cells, the addition of cholesterol does not induce higher internalization, reflecting that AMPARs are docked at the PSD and stimulation is indispensable for the internalization machinery to become activated. Supporting our theory, we now show that the addition of cholesterol to 30DIV neurons in control conditions does not result in increased lateral mobility of confined receptors (D_i). This is now remarked in the results section (underlined on page 11) and included in the discussion (see underlined text, page 17).

Additionally, we now clarify the meaning of $n=3$ in Figure 3 (underlined on page 33) and clarify that the effect of cholesterol on receptor internalization occurs exclusively on neurons after glutamate stimulation.

Our experiments were designed to analyze if the cholesterol decrease in neuronal membranes is responsible for the impairment observed in all the mechanisms underlying LTD *in vivo*. Thus, our approaches were either to analyze the effect of removing cholesterol from cells with normal content of this lipid or adding cholesterol to cells with low content to rescue the normal function. However, if cholesterol addition to young neurons would have any remarkable effect on AMPAR endocytosis, this effect would be reflected by a reduced or enhanced LTD, a phenomenon that was not observed in our electrophysiological data *in vivo* or *ex-vivo*. Furthermore, as we explain now in the text, cholesterol addition to 15DIV neurons does not affect the lateral diffusion of confined AMPARs after stimulation, a phenomenon that precedes endocytosis.

Comment 5. 15 DIV cells should also be included in the experiments on cholesterol effects on lateral mobility of GluR2 (fig.5).

Answer: the data corresponding to lateral mobility in 15 days *in vitro* neurons is now included (see new Supporting Information Figure 4B and underlined in page 11).

Comment 6. For LTD measurements, cholesterol has been applied 30 min before to 30 min after NMDA stimulation. To test whether cholesterol exerts sustained effects of LTD, slices should be pre-treated with cholesterol before NMDA stimulation and then analyzed in absence of cholesterol. In turn, cholesterol should only be added after or together with NMDA stimulation. These variations might allow distinguish between sustained and acute effects of cholesterol on LTD.

Answer: in those experiments, acute slices need to be pretreated at least 30 min with the cholesterol solution to restore the normal cholesterol levels observed in young individuals.

A treatment with cholesterol only during the 5 min of NMDA would not be sufficient for the lipid to be incorporated in the neurons. The addition of cholesterol after NMDA induction would not work for the same reason. If LTD is induced in old slices without any cholesterol pretreatment, then, the stimulus won't be transduced and the LTD will be lost before the cholesterol is incorporated.

We could eliminate the treatment with cholesterol 30 min after induction to show the sustained effect of cholesterol. However, this sustained effect can be observed in the Figure 1A since long-term depression is still observed at 60min after NMDA application (30 min after the cholesterol solution was removed).

Comment 7. Effects of cholesterol extraction by MCD on LTD are only shown for 10 M mice (fig.6b). Does MCD treatment affect LTD in younger (2M) mice? Similar question arises for the experiments with rats (fig.7 and 8). Does cholesterol application affect LTD and memory in younger animals? This appears important to assess whether cholesterol application generally strengthens LTD or selectively reverts deficits in aged mice with gradual loss of cholesterol.

Answer: We provide with new data showing that cholesterol addition to slices from young animals does not result in any improvement in LTD (see new Figure 6A). In addition, we provide with new data showing that addition of cholesterol directly in the hippocampus of young rats does not improve LTD (see new Figure 7B). Altogether these data indicate that cholesterol only has a beneficial effect when acting on cholesterol pauperized background.

Additional points:

- the error bars should also be provided for controls in fig. 1.*
- information on the number of experiments and samples, cells should be provided for each quantification (fig. 1a,b,c; fig.2; fig.3c, fig.5;).*
- reference 'Jurado et al., 2010' cited on p.6 is missing in the reference list*
- the ms should be carefully checked for phasing and grammatical errors.*

Answer:

The data presented in the Figure 1 show the fold change of the signal intensity obtained in each condition respect to its control, thus the controls from different experiments have always the value of 1 (and the error is 0). This is the only way to compare the values obtained in different western blots, where each western blot is a different experiment.

The information required about number of experiments and samples in now indicated in the each figure legend.

All the other queries have been addressed, including native English proof-reading and editing.

Referee #3.

Comment:

This is a very interesting study implicating lowered cholesterol content in the impairment of LTD during aging. For the most part the experiments are well designed and the results convincing. I do have, however, some **reservation about the relevance of the studies concerning the internalization and mobility of the AMPA receptors**. Those were done in cell cultures where LTD was induced with glutamate and monitored by imaging GluR2; in contrast, the evaluation of LTD in the aged tissue was done in slices, using NMDA and monitored as changes in synaptic response. **First, the loss of internalization observed in the 30 days in the life of a culture cell it is not a model for the loss of LTD that occurs in the course on months. Second, glutamate (as used in the culture setting) can induce two different form of LTD: NMDAR-dependent LTD, which is not dependent on GLuR2, and mGluR-dependent LTD, which does depend on GluR2. It must be noted, however, that even without those experiments, the case for a crucial role of lowered cholesterol in the impairment of LTD during aging is still strong. Perhaps a simple solution will be to remove figures 3 and 4.**

Answer: We totally agree with this referee in that 30 days *in vitro* hippocampal neurons cannot be taken as the counterpart of hippocampal neurons in the brains of 20 month-old mice. On the other hand, we recognize that neurons *in vitro* are a valid tool to explore and validate mechanisms of neuronal function. In our case, the use of neurons *in vitro* was justified to add cell biological strength to the *in vivo* and *ex-vivo* data. We use the cultures as a tool to study AMPAR dynamics in neuronal populations with different cholesterol content. In support of this, the data in Figures 3 and 4 show that only neurons that have lost 25% cholesterol *in vitro* (30 days) had cell biological deficits in processes that are required for LTD *in vivo*, namely reduced lateral diffusion and reduced internalization after excitatory neurotransmitter addition. Moreover, the data in Figures 3 and 4 show that only the neurons at 30 days *in vitro* responded to the benefits of cholesterol addition, increasing receptor lateral mobility and increasing receptor internalization.

With respect to the glutamate treatment *in vitro*, we agree with the reviewer that glutamate addition to neuronal cultures can not be considered that induce LTD. However we chose bath application of glutamate to stimulate neurons in culture because it was previously demonstrated that in hippocampal cultures this ligand induces AMPAR endocytosis by a mechanism shared by LTD. Furthermore, it was also shown (using the same single molecule tracking experiments), that glutamate incubation increased the diffusion coefficient of synaptic AMPARs in mature young neurons in the time-window required for these experiments.

In other words, the *in vitro* data provide cell biological support to the *ex-vivo* and *in vivo* electrophysiological and biochemical data, moving us to ask this reviewer to allow us to keep these two figures. In response to the reviewer's concerns, we have clarified in the text that we used neurons *in vitro* as cell biology validation tool and not as a model of aging (underlined text, page 8 and 9).

The citations reporting a loss of LTD during aging need some working. There are changes of LTD throughout the lifespan and at different times. Some are maturational processes that occur within the few postnatal weeks (Dudek and Bear, Bear and Abraham, Milner, Kemp), other occur during middle age or after (Amhed et al, Lee et al).

Answer: we have now only left in the text the references concerning LTD decay in the aged (underlined text, page 4).

Other points

The reference Jurado et al missing

Most of the acronyms are introduced without definition, making it difficult for a non-specialist. For example p-Akt was brought to the introduction without any warning.

Answer: reference was corrected and acronyms clarified.

Referee #3 (Remarks): It will be a very nice study if cell culture experiments are removed or better justified

Answer: Now cell culture experiments are justified according to this reviewer's concerns

Thank you for the submission of your revised manuscript to EMBO Molecular Medicine. We have finally received the enclosed report from the referee who was asked to re-assess it. As you will see the reviewer is now globally supportive of publication, although the involvement of MARCKS in cholesterol induced changes are again under scrutiny. Would you have data at hand to answer these concerns, I would encourage you to add them to the manuscript. However, I would suggest not removing this part of the work from the manuscript.

I look forward to seeing a revised form of your manuscript as soon as possible.

***** Reviewer's comments *****

Referee #2 (Remarks):

In the revised version of this manuscript, authors addressed most points of the reviewer by additional experiments, and corresponding revision of figures and text. The manuscript clearly improved and results are very interesting. However, I still feel that the functional involvement of MARCKS in the cholesterol induced changes is not well documented. Authors first use 15DIV primary cultures to show effects of cholesterol depletion on membrane association of MARCKS (fig.1D). However, pAKT and AKT levels have not been analyzed in these experiments. In fig. 1E, brain slices of 4M mice were used to show that supplementation with cholesterol had no effect on membrane association of MARCKS. It would be more comprehensive to show the effects of cholesterol depletion AND cholesterol addition in both models (primary cultures and slices), including the analysis of membrane distribution of MARCKS and levels of pAKT/AKT. As mentioned in the first review, it would also be interesting to analyze the changes of MARCKS upon cholesterol modulation (rescue) in 'old' neurons/slices. Authors also included new experiments with shRNA mediated knockdown of MARCKS (in 15 DIV neurons) and show increased pAKT levels (supplementary fig. 1). Cholesterol extraction in shRNA knockdown cells also increased pAKT. These data appear very descriptive. To test a functional connection of these observations, the effects of cholesterol depletion on pAKT in shRNA knockdown cells need to be compared to that in control (non-targeting shRNA) cells. The most interesting point of the study is the beneficial effect of cholesterol supplementation in 'aged' cellular and animal models. This is very well supported by the data and comprehensively addressed. Thus, I don't see a benefit of including the quite descriptive and rather preliminary data on the involvement of MARCKS in the study. Authors might consider to remove the data on MARCKS from the study or strengthen this issue by additional experiments mentioned above.

Other points for revision have been well addressed by the authors.

Additional Author Correspondence

11 April 2014

Many thanks for your letter and your encouraging words, supportive of our work. I am now organizing a meeting with Mauricio (the first author) to see whether we have some additional data worth including (on Marcks) and to comply with all other requests. Regarding this reviewer's low level of "satisfaction" with our MARCKS-Akt data, I would like to comment that the relationship between MARCKS membrane levels and activity of Akt was addressed in a previous publication from my laboratory (Trovú et al., Nature Neuroscience 2013). In that work we showed that when MARCKS is lost it leads to higher active Akt and when MARCKS levels are increased the levels of Akt do not change. In other words, the mechanisms downstream of MARCKS was part of our previous work. The current work for EMM is on the mechanism upstream MARCKS loss. Given that both series of studies were in the aged context, we did not re-add to the current work the downstream consequences of losing MARCKS, that would have been redundant. To me, a good

compromise would be to better explain what happens downstream of MARCKS when this protein is lost: would you be satisfied by this ?

Naturally, we will comply with all other requests. Hopefully soon we will be submitting the final corrected version.

Additional Editorial Correspondence

14 April 2014

Thank you very much for your e-mail. I fully agree that in case you don't find additional data to reply to the referee's comments, a better explanation of MARCKS downstream events would be a good compromise indeed.

I am looking forward to receiving your final manuscript.